# Barbed arrow-like structure membrane with ultra-high rectification coefficient enables ultra-fast, highly-sensitive lateral-flow assay of cTnI

Juanhua Li[1,2,5], Yiren Liu ⬥[1,2,5], Tianyu Wu[1,2], Zihan Xiao ⬥[1,2], Jianhang Du[3], Hongrui Liang[1,2], Cuiping Zhou[4] & Jianhua Zhou ⬥[1,2] ✉

Acute myocardial infarction (AMI) has become a public health disease threatening public life safety due to its high mortality. The lateral-flow assay (LFA) of a typical cardiac biomarker, troponin I (cTnI), is essential for the timely warnings of AMI. However, it is a challenge to achieve an ultra-fast and highly-sensitive assay for cTnI (hs-cTnI) using current LFA, due to the limited performance of chromatographic membranes. Here, we propose a barbed arrow-like structure membrane (BAS Mem), which enables the unidirectional, fast flow and low-residual of liquid. The liquid is rectified through the forces generated by the sidewalls of the barbed arrow-like grooves. The rectification coefficient of liquid flow on BAS Mem is 14.5 (highest to date). Using BAS Mem to replace the conventional chromatographic membrane, we prepare batches of lateral-flow strips and achieve LFA of cTnI within 240 s, with a limit of detection of 1.97 ng mL$^{-1}$. The lateral-flow strips exhibit a specificity of 100%, a sensitivity of 93.3% in detecting 25 samples of suspected AMI patients. The lateral-flow strips show great performance in providing reliable results for clinical diagnosis, with the potential to provide early warnings for AMI.

Acute myocardial infarction (AMI) is a global cardiovascular disease that endangers human life's health. Patients may experience rapid deterioration within hours of symptom onset, such as chest pain and shortness of breath, which can potentially lead to sudden death[1,2]. A hallmark of AMI is myocardial injury, primarily resulting from myocardial ischemia (a sudden reduction or cessation of coronary artery blood flow), which leads to the abnormal release of cardiac biomarkers into the peripheral blood[3,4]. Specifically, the abnormal increase in cardiac troponin I (cTnI) content in the blood has become a highly specific indication for the early diagnosis and prognosis assessment of

AMI patients[4–6]. Consequently, there is an urgent need for a fast and highly sensitive assay of cTnI (hs-cTnI assay), which meets the demand for rapid and accurate early detection of AMI, and helps to prompt early warnings and save patients' lives in the early stages of onset[7,8].

Lateral-flow assay (LFA) is a typical point-of-care testing (POCT) method, offering results within 30 min and finding extensive application in rapid assays[9–11]. Here, the nanogold LFA, the most widely-used LFA, for the sandwich immunoassay of cTnI is used as a typical example to introduce the antigen-antibody binding in current LFA. When a sample solution containing cTnI (e.g., peripheral blood serum)

[1]School of Biomedical Engineering, Shenzhen Campus of Sun Yat-sen University, Shenzhen, Guangdong, China. [2]Key Laboratory of Sensing Technology and Biomedical Instruments of Guangdong Province, School of Biomedical Engineering, Sun Yat-sen University, Guangzhou, Guangdong, China. [3]Guangdong Innovative Engineering and Technology Research Center for Assisted Circulation, the Eighth Affiliated Hospital of Sun Yat-sen University, Shenzhen, Guangdong, China. [4]Department of Emergency, Nanfang Hospital, Southern Medical University, Guangzhou, Guangdong, China. [5]These authors contributed equally: Juanhua Li, Yiren Liu. ✉e-mail: zhoujh33@mail.sysu.edu.cn

is loaded onto the sample pad of lateral-flow strip, the cTnI would bind with nanogold-labeled antibodies, forming "cTnI-antibody-nanogold" as the detectable complexes. The detectable complexes, antigen, and nanogold-labeled antibodies, which are termed multi-complexes for short, would be released from the conjugate pad. Then the multi-complexes would migrate along the lateral-flow membrane due to capillary action, and would be captured selectively by the antibodies coated at the test line (T line) and by the antibodies coated at the control line (C line)[12,13]. In this method, it is evidently crucial for lateral-flow membrane to drive the flow of the sample solutions through passive microfluidics, and to selectively capture the complexes carried in the solution[14]. Currently, most lateral-flow membranes are made of micron-diameter fiber filaments and enriched with numerous internal tiny pores, and hereafter termed fiber-based chromatographic membranes. While adjusting the hydrophobicity and hydrophilicity of the fibers can achieve the flow of the sample solution and the immobilization of antibodies, such modifications also limit the performance of LFA, particularly increasing the detection time and decreasing the signal-to-noise ratio (SNR) of the assay[15,16]. 1) The slow flow of sample solution in fiber-based chromatographic membranes (typically under $0.5 \, \text{mm s}^{-1}$) prolongs the detection time of LFA to 15–30 min. 2) The abundant pores in the fiber-based chromatographic membranes would trap the sample solution, resulting in the increased non-specific adsorption of detectable complexes and the trap of multi-complexes. 3) The residual detectable complexes in the area of fiber-based chromatographic membranes aside from T line and C line would reduce the amounts of that reaching to the T line and C line, resulting in the decline of the detection signal. 4) The residual muti-complexes (specifically refers to the nanogold-labeled antibodies and "antigen-antibody-nanogold") at T line and C line would increase the background noise, resulting in a suboptimal SNR and even a fake positive result. Addressing these limitations necessitates a new-generation lateral-flow membrane that provides the faster velocity of solution flow and lower residues of sample solution and helps to improve the performance of LFA[17].

In this study, we propose a barbed arrow-like structure membrane (BAS Mem) that promotes unidirectional and fast flow of liquid. The surface of BAS Mem features interconnected barbed arrow-like grooves head to tail, and the grooves are bounded by sidewalls. The sidewalls of the barbed arrow-like grooves on BAS Mem are the key to driving the fast flow of liquid and restricting the flow of liquid in the opposite direction. The BAS Mem boasts an ultra-high rectification coefficient of 14.5 and an ultra-fast flow velocity of liquid that is about 23 times higher than that of the nitrocellulose membrane (NC Mem). We further use the BAS Mem as the lateral-flow membrane of the lateral-flow strip to investigate its detection performance for LFA. The results indicate employing the BAS Mem in lateral-flow strips can reduce the detection time and improve the SNR of LFA. We then prepare batches of lateral-flow strips using the BAS Mem, and develop a nanogold LFA for an ultra-fast hs-cTnI assay. The LFA based on proposed lateral-flow strips can provide results in just 240 s with the limit of detection (LOD) of $1.97 \, \text{pg mL}^{-1}$. Across the detection of serum samples of 25 suspected AMI patients, the assays demonstrate 100% specificity and 93.3% sensitivity. According to the laboratory medical practice guidelines of the National Academy of Clinical Biochemistry (NACB) and the International Federation of Clinical Chemistry (IFCC), this method meets the requirement for the hs-cTnI assay[18], so that it is expected to diagnose AMI timely and early, thereby save patients' lives in this way.

## Results

### Design of BAS Mem and principle for unidirectional transport

We designed the surface structure of the BAS Mem (Fig. 1A) which could drive the flow of water unidirectionally, investigated the principle of unidirectional flow of liquid on the membrane by analyzing the forces acting on the meniscuses (Fig. 1B, C). The BAS Mem was applied as a lateral-flow membrane to construct the lateral-flow strips and develop a nanogold LFA for the ultra-fast hs-cTnI assay, which helped to diagnose AMI (Fig. 1D, E). As shown in Fig. 1A–i, the surface of BAS Mem featured parallel and equidistant microchannels. The microchannels were comprised of interconnected barbed arrow-like grooves head to tail, which termed as "the structure unit". The structure units were bounded by a set of axisymmetrically distributed long-arc sidewalls and short-arc sidewalls. The long-arc sidewalls tangentially met the short-arc sidewalls at a 0° angle, creating the sharp corners where the sidewalls intersected. These unique structures on the BAS Mem were prepared through techniques of laser carving, casting and hot embossing. First, we laser-carved microchannels onto an aluminum sheet to produce a metal mold. We then poured the polydimethylsiloxane (PDMS) prepolymer onto the mold, heating it to obtain the PDMS master. Subsequently, we employed a hot compressor to emboss the pattern of the PDMS master onto a polymer membrane, such as the high-density polyethylene (HDPE) membrane. After applying a hydrophilic coating to the surface pattern of the membrane, the preparation of BAS Mem was complete.

To investigate the principle of unidirectional flow of liquid on the BAS Mem, we marked four variable parameters to define the shape of a structure unit (Fig. 1Aiii–iv). Here, α denoted the degrees of the short arc on the short-arc sidewalls, and β denoted the degrees of the long arc on the long-arc sidewalls, respectively. W denoted the shortest width between the long-arc sidewalls within a structure unit. H denoted the heights of the sidewalls. The analysis of forces on meniscuses was discussed (Fig. 1B, C). Here, the flow direction of liquid from the head of a structure unit to its tail was defined as the "forward direction", while the reverse one was noted as the "backward direction of liquid flowing". The flow of liquid on the BAS Mem was influenced by forces from both the undersides and sidewalls of the structure units. As liquid flowed backward to the sharp corners, the continually arriving liquid formed a convex meniscus at the head of the structure unit due to the constraint of the sidewall contact line (Fig. 1B). As per Gibbs inequality[19], varying intrinsic contact angles on membranes may result in different flow behavior of liquid. We discussed three cases when there were different value relationships between the angle α and the contact angle θ (Supplementary Fig. 1). One interesting finding was that the forces generated by sidewalls were always in the opposite direction to the flow direction of liquid. On account of the resistance characteristics of flowing, the force generated by the sidewall was denoted as $\mathbf{F}_{r1}$ (Supplementary Discussion 1). In contrast, the force generated by the underside was denoted as $\mathbf{F}_{ud_1}$, which always had the same direction as the flow direction of liquid (Fig. 1B)[20,21]. Therefore, as liquid flowed backwards and pinned at the sharp corner, the total force acting on the convex meniscus, $\mathbf{F}_{tp}$, could express as (1).

$$\mathbf{F}_{tp} = 2\mathbf{F}_{r1} - \mathbf{F}_{ud_1} = \sigma H^2[1 - \sin(\alpha - \theta)] - \sigma W \cos\theta \qquad (1)$$

Here, $\sigma$ represented the coefficient of surface tension. $\theta$ was the intrinsic contact angle.

As water flowed forward on the BAS Mem, the sidewalls generated forces in different directions (Fig. 1C). $\mathbf{F}_{sd_2}$ represented the driving forces of liquid flowing, while $\mathbf{F}_{r2}$ represented the resistance forces. They could be obtained by decomposing $\mathbf{F}_1$ and $\mathbf{F}_2$ (Supplementary Fig. 2 and Supplementary Discussion 1). And the force generated by the underside was still in the same direction as liquid flowed, which was denoted as $\mathbf{F}_{ud_2}$. Therefore, as liquid flow forward, the total force acting on concave meniscus, $\mathbf{F}_{ts}$, could express as (2).

$$\mathbf{F}_{ts} = \mathbf{F}_{ud_2} + 2\mathbf{F}_{sd_2} - 2\mathbf{F}_{r2} = \sigma l \cos\theta + 2\sigma H \cos\theta \cos\gamma - 2\sigma H \sin\gamma \quad (2)$$

Here, the line of contact between the concave meniscus and the underside was approximated as an arc. $l$ represented the chord length

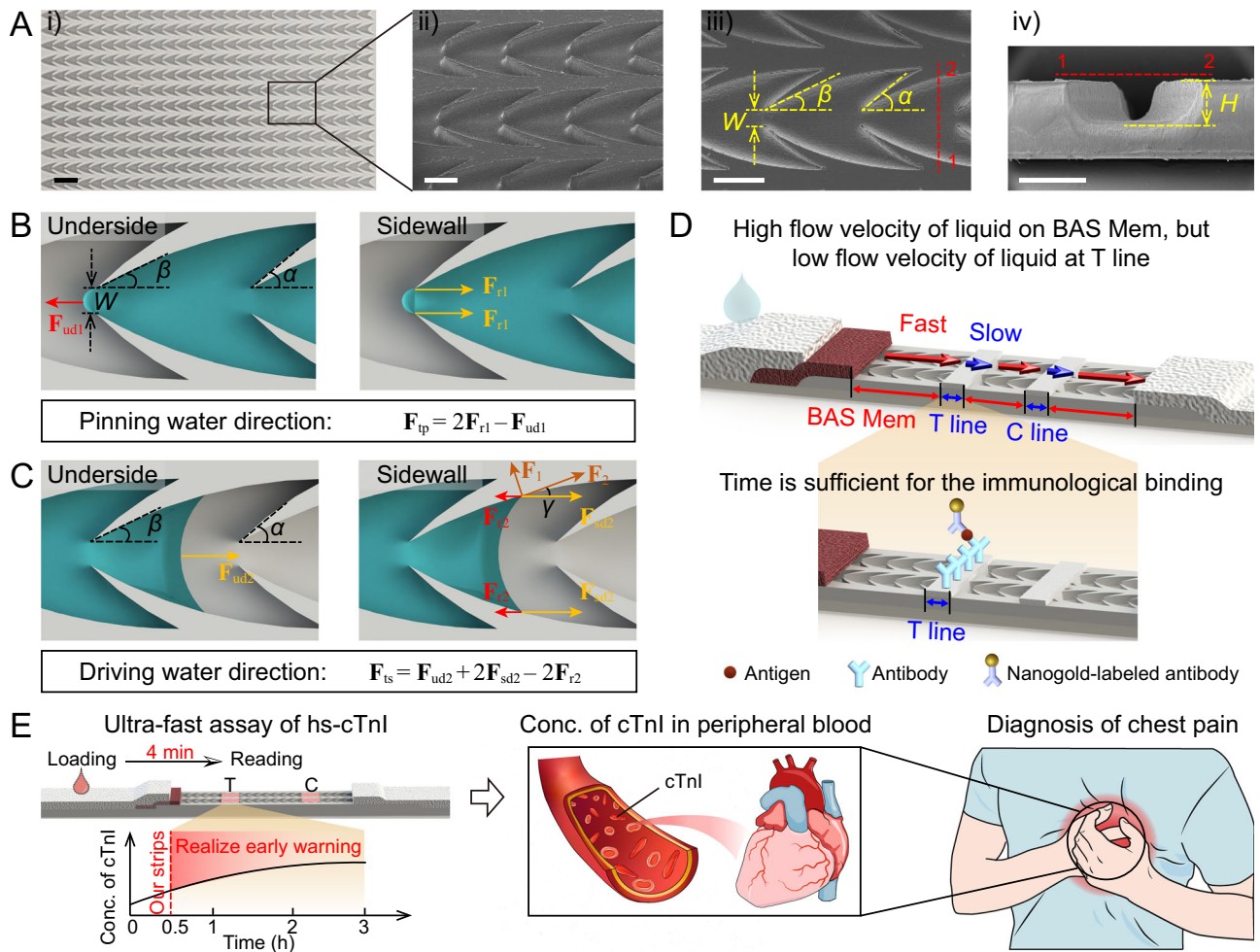

**Fig. 1 | Design and structure characterization of the BAS Mem, the principle of unidirectional flow of liquid on BAS Mem, the flow of liquid on a mimic strip and the application of BAS Mem to construct lateral-flow strips for ultra-fast hs-cTnI assay.** A Design and structure characterization of the BAS Mem, images representative of 3 experiments. A -i) Optical image of the BAS Mem. Scale bar = 2 mm; A-ii) Magnified SEM image of the BAS Mem. Scale bar = 500 μm; A-iii) Top-SEM view of the BAS Mem. Here, α denoted the degrees of the short arc on the short-arc sidewalls, and β denoted the degrees of the long arc on the long-arc sidewalls, respectively. W denoted the shortest width between the long-arc side-walls within a structure unit. Scale bar = 500 μm; A-iv) SEM image of a cross-sectional view of the BAS Mem. H denoted the heights of the sidewalls. Scale bar = 500 μm. **B** Schematic drawings indicate the analysis of forces on the convex meniscuses when liquid was pinned on the BAS Mem, including the forces gener-ated by the underside (left) and that by sidewall (right) respectively. **C** Schematic drawings indicate the analysis of forces on the concave meniscuses when water was spreading on the BAS Mem, including the forces generated by the underside (left) and that by sidewall (right) respectively. **D** Conceptual graph showing the char-acteristics of the lateral-flow strip made of BAS Mem, highlighting that the flow velocity is high on BAS Mem but low at T line, and the time is sufficient for the immunological binding in T line. **E** Application of BAS Mem for ultra-fast lateral-flow assay of hs-cTnI. The schematic drawing shows the process of detecting cTnI in peripheral blood within 4 min using BAS Mem-based lateral-flow strips for diag-nosis of chest pain that occurs in humans.

of the arc, satisfying $l \in (W,D)$. $\gamma$ represented the tangential angle at the endpoint of this arc, satisfying $\gamma \in (0,\beta)$. Overall, we found the fact that the sidewalls provided resistance to the flow as the liquid flowed backward, which was the key to generating the unidirectional flow of liquid on the BAS Mem.

We also investigated the performance of BAS Mem with different structure parameters in carrying liquid, in terms of directionality and flow velocity of liquid. Here, we defined the minimum spacings between neighboring microchannels as "S" (Supplementary Fig. 3). The membranes with different structure parameters, as detailed in Sup-plementary Tables 1–4, were fabricated, and the scanning electron microscopy (SEM) images of their surface structures were exhibited in Supplementary Fig. 4. We dropped red ink (~15 μL) to these mem-branes, and recorded lengths of red ink liquid flowed to both direc-tions (Supplementary Fig. 5). The rectification coefficient of liquid flow

on membranes, which was termed "k", could express as (3).

$$k = \frac{L_s}{L_p} \qquad (3)$$

Herein, $L_s$ represented the lengths of liquid forward flowing, and $L_p$ represented the lengths of liquid backward flowing. The rectifica-tion coefficient has broadly used to reflect the ability of membranes to drive liquid unidirectionally[22]. The higher value of k was, the stronger ability to drive liquid unidirectionally the BAS Mem had. There was a high value of k when the structure parameters were φ = 15°, S = 0.24 mm, H = 0.32 mm, W = 0.16 mm. We also evaluated the flow velocity of liquid on different membranes. As shown in Supplementary Fig. 6, the curves corresponding to φ = 15°, S = 0.24 mm, H = 0.32 mm, W = 0.20 mm reached the inflection point earlier, indicating that the

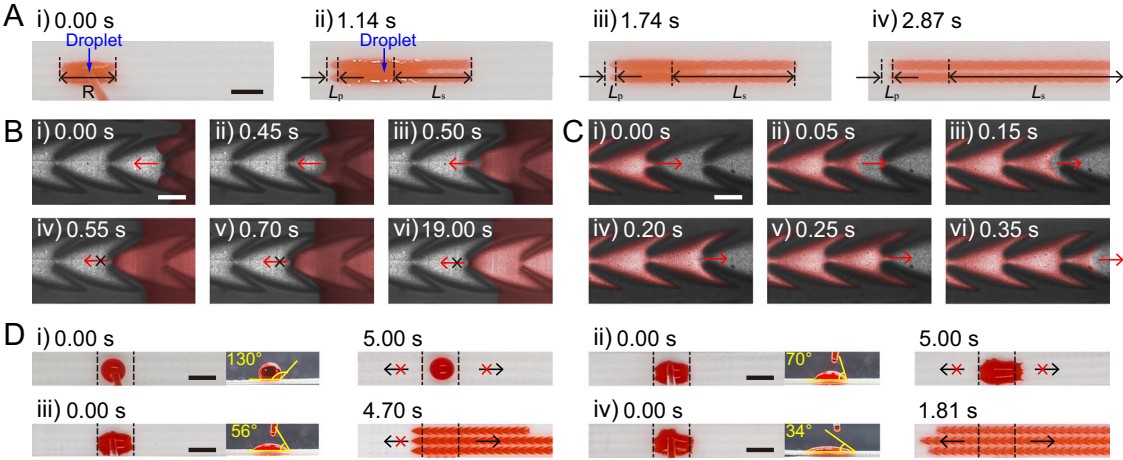

**Fig. 2 | Unidirectional flow behavior of liquid on BAS Mem. A** Optical images of time-dependent unidirectional flow behavior of liquid on BAS Mem. The droplet could unidirectionally spread. Scale bar = 3 mm. **B** High-speed digital images show that the liquid pinned within a single structure unit on BAS Mem, images representative of 3 experiments. Scale bar = 500 μm. **C** High-speed digital images show the spreading of liquid in a single structure unit on BAS Mem, images representative of 3 experiments. Scale bar = 500 μm. **D** Optical images show the flow behavior of liquid on different BAS Mems which were modified to display different surface contact angle. Scale bar = 3 mm.

BAS Mem with these parameters could drive liquid fast. Overall, the BAS Mem with structural parameters φ = 15° (α = 45°, β = 30°), S = 0.24 mm, H = 0.32 mm, W = 0.16 mm was preferentially selected, considering the performance of both the rectification coefficient and flow velocity of liquid.

This design could be processed on various materials, such as polymethyl methacrylate (PMMA) and polypropylene (PP) (Supplementary Fig. 7). Through the preparation technology described above, we are able to avoid the use of fiber filaments, which are generally hydrophobic, negatively charged fibers (like those of nitrocellulose) and can non-specifically adsorb multi-complexes. We used the BAS Mem with different materials as raw materials to transport the sample solution of FITC-labeled albumin from bovine serum (F-BSA) (Supplementary Fig. 8). The results showed that the non-specific adsorption of proteins on BAS Mem was lower than that on the NC Mem. Additionally, multiple batches of BAS Mem were also prepared using HDPE membranes (Supplementary Fig. 9), and used to drive liquid (Supplementary Fig. 10 and Supplementary Movie 1 and 2), confirming consistent performance across batches through the preparation technology we used. The long-term hydrophilicity of the surface of BAS Mem had also been confirmed experimentally. The prepared BAS Mem maintained its hydrophilic properties consistently, and could transport the liquid effectively after a storage period of six months (Supplementary Fig. 11). Due to its capability to drive liquid unidirectionally, BAS Mem could be used to construct lateral-flow strips for LFA. Figure 1D conceptually illustrates the flow of liquid on the lateral-flow strips. It showed that the flow velocity was high on the BAS Mem of our lateral-flow strip, whereas it was lower at the T line. The reduced flow velocity at the T line resulted in a longer residence time of the liquid at this location. This extended residence time was crucial, as it allowed sufficient time for the immunological binding between the "antigen-antibody-nanogold" complex and the antibody immobilized in the T line. Using the BAS Mem as a key component, the lateral-flow strips were constructed to develop a nanogold LFA, which was available for the ultra-fast hs-cTnI assay. It only takes 4 min to detect the concentrations of cTnI in peripheral blood using the lateral-flow strips, realizing the early warning of AMI (Fig. 1E) and helping to diagnose symptoms of AMI, such as chest pain. It assists the earlier emergency management and treatment for AMI and, therefore, lowers the rate of mortality.

## Unidirectional flow behavior of liquid on BAS Mem

Based on the theoretical force analysis of meniscuses shown in Fig. 1B–D, we conducted further experiments on the unidirectional flow behavior of liquid on the BAS Mem, using red ink as a visual aid. As shown in Fig. 2A and Supplementary Movie 3, the red ink (-10 μL) on the BAS Mem continuously flowed forward for a long distance, while stopped flowing after a short distance backward. We used "$L_s$" to denote the length of forward flow and "$L_p$" for the length of backward flow. In the initial 1.14 s, the red ink exhibited a pronounced forward flow ($L_s$ = 7.2 mm) and a slight backward flow ($L_p$ = 1.0 mm). After 1.14 s, the red ink only flowed forward, causing a continuous increase in $L_s$ while $L_p$ remained unchanged. By 2.87 s, the ink had covered -15.0 mm forward and a mere 1.0 mm backward on the BAS Mem. The experimental results suggested that the liquid could flow unidirectionally on BAS Mem.

We subsequently investigated the microscopic flow behavior of the liquid in the microchannels on the BAS Mem. We captured both the forward and backward flow of liquid in the microchannels by using a high-speed camera (refer to Supplementary Movie 4, 5). As shown in Fig. 2B, the liquid flowed backwards to the tip of the structure unit after 0.50 s, with the meniscus pinned at the sharp corners, the BAS Mem halted further backward flow of liquid. In contrast, the liquid maintained its forward momentum on BAS Mem, filling one structure unit in just 0.25 s before advancing to the subsequent one (Fig. 2C). We also used Ansys Fluent 2021 (Fluent, USA) to simulate the flow behavior of liquid in a structure unit (Supplementary Fig. 12). The simulate results revealed that the droplet expanded within the structure unit, advanced to the forward next, but was pinned at sharp corners when moving backward. This simulation aligned well with our experimental findings. Both the experimental observation and the simulation supported the theoretical analysis presented in Fig. 1B, C. The sidewall-generated resistance forces acting on the meniscuses inhibited the backward flow of liquid, while the combined driving forces from the sidewalls and undersides drove the liquid to flow forward. In consequence, the liquid could flow unidirectionally in microchannels on the BAS Mem.

We further studied the conditions for the unidirectional flow of liquid on BAS Mem. We first performed theoretical calculations. In view of the force analysis on meniscuses, a unidirectional flow of liquid was achievable when both $\mathbf{F}_{tp} > 0$ and $\mathbf{F}_{td} > 0$ were satisfied (Supplementary Discussion 2). Experimentally, we observed the flow behavior

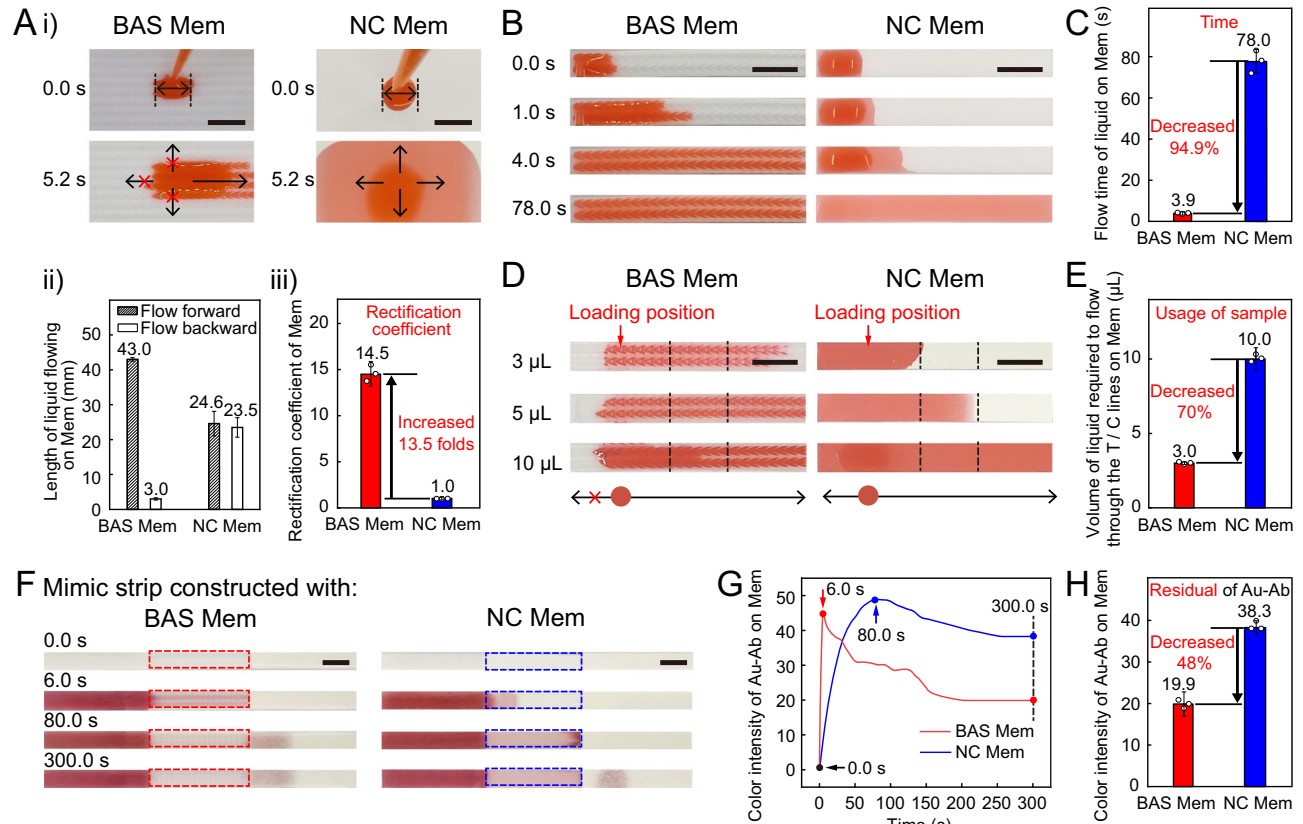

**Fig. 3 | Characterization of the performance of BAS Mem when they were used to drive liquid and sample solution, including flow directionality of liquid, flow time of liquid, utilization rate of liquid, and residual sample solution on the membrane. A** Characterization of flow behavior of liquid on two types of membranes. A-i) Optical images show the flow behavior of liquid on two types of membranes. Scale bar = 5 mm.; A-ii) Comparison histogram shows the lengths of liquid flowing on two types of membranes; A-iii) Comparison histogram of the rectification coefficients between two types of membranes. **B** Optical images show the time-dependent flow of liquid on two types of membranes. Scale bar = 5 mm. **C** Comparison histogram shows the time of liquid flowing on two types of

membranes. **D** Optical images show the flow of liquid in different volumes on two types of membranes. The black dashed lines marked in the image indicate the location of practical T and C lines. Scale bar = 5 mm. **E** Comparison histogram shows the lengths of 5 μL of liquid flowing on two types of membranes. **F** Optical images show the time-dependent flow of sample solution on two types of membranes, when the sample solution of Au-Ab was transported using mimic strips. Scale bar = 5 mm. **G** Curves show the time-dependent color intensity of Au-Ab on membranes. **H** Comparison histogram show the color intensity of Au-Ab that remained on BAS Mem and NC Mem (at 300 s). Data are presented as mean ± SD in **A**-ii ($n = 3$), **A**-iii ($n = 3$), **C** ($n = 3$), **E** ($n = 3$), **H** ($n = 3$, independent replicates).

of liquid on surfaces of BAS Mem that vary in contact angles ($\theta$), as showcased in Supplementary Movie 6 a throughout the manuscript and Fig. 2D. We achieved membranes with distinct values of $\theta$, by treating the surfaces of BAS Mem with oxygen plasma for different periods of time[23]. After dropping 10 μL of liquid into these modified membranes, we observed the following: at $\theta = 130°$, the liquid remained stationary (Fig. D-i); at $\theta = 70°$, the liquid exhibited limited flow (Fig. D-ii); at $\theta = 56°$, the liquid flowed unidirectionally on the membrane (Fig. D-iii); and at $\theta = 34°$, the liquid flowed in both directions on the membrane (Fig. D-iv). Complementing these observations, we simulated the flow behavior of liquid on the membranes with surface contact angles of 70°, 56°, and 34° (Supplementary Fig. 13). The simulation outcomes were consistent with experimental findings, indicating that the BAS Mem drove liquid unidirectionally when the surface contact angle was 48° < $\theta$ < 64°.

**The performance of BAS Mem for LFA**
Next, we conducted experiments to assess the performance of the LFA using BAS Mem, including the detection time and SNR. We compared BAS Mem with the prevalent NC Mem, a typical example of fiber-based chromatographic membranes, in terms of the flow directionality of liquid, flow time of liquid, utilization rate of liquid, and residual of sample solution on the membrane.

First, we investigated the directionality of liquid flow on the membranes. We dropped 30 μL of liquid (i.e., red ink) onto the middle part of both membranes (Fig. 3A–i and Supplementary Movie 7, 8). After 5.23 s, the liquid only flowed from the head to the tail of the structure unit on BAS Mem, while the liquid flowed in all directions freely in NC Mem. Figure 3A–ii presents the bar graph comparing the length of forward and backward liquid flowing in millimeters. The liquid on the BAS Mem exhibited a forward flow of 43.0 mm, and a backward flow of merely 3.0 mm. In contrast, the lengths of liquid flow in both directions on the NC Mem were relatively similar. As shown in Fig. 3A–iii, the rectification coefficient for BAS Mem reached 14.5, which is 13.5 times higher than that of the NC Mem (1.02). This measured value of rectification coefficient of BAS Mem is higher than that of others previously reported[24–30].

Then we investigated the flow time of liquid on membranes. We dropped 10 μL of liquid (i.e., red ink) onto both BAS Mem and NC Mem, and observed the respective flow behaviors of liquid on these membranes (Supplementary Movie 9, 10). As shown in Fig. 3B, the liquid on BAS Mem was able to spread 6.8 mm at 1.0 s, while the liquid could only spread 1.2 mm in NC Mem. The BAS Mem facilitated a 20.0 mm spread in ~4.0 s, whereas the NC Mem required 78.0 s for the same spread (Fig. 3B). The flow time of liquid on the BAS Mem ~93.5% lower than that in the NC Mem (Fig. 3C), indicating a shorter flow time of

liquid on the BAS Mem compared to the NC Mem. Further calculations revealed that the average flow velocity on the BAS Mem was 5.48 mm s⁻¹, which was ~23 times faster than in the NC Mem (0.23 mm s⁻¹) (Supplementary Fig. 14). These results suggested the BAS Mem enabled shorter flow time of liquid, comparing to the NC Mem. Consequently, the BAS Mem has the potential to shorten the time of sample solution taken to flow to the T and C lines on lateral-flow strips and has contributed to the reduction of detection time in LFA.

We further investigated the utilization rate of the sample solution when flowing on BAS Mem. The ultra-high rectification coefficient of BAS Mem indicated that the BAS Mem remained a strong restriction for the flow direction of liquid on membrane, which avoided the liquid losing in the other flow directions of BAS Mem. In this way, only a small volume of sample solution would be sufficient to flow through the entire BAS Mem. To experimentally verify this, we first cut the two types of membranes, then dropped an identical volume of red ink to each membrane and observed the flow (Supplementary Movie 11, 12). As could be seen in Fig. 3D, a mere 3 µL of red ink on the BAS Mem sufficed to flow through the T and C lines. Conversely, NC Mem necessitated a minimum of 10 µL to achieve the same. The usage of sample on the BAS Mem was ~70% lower than that in the NC Mem (Fig. 3E), indicating a higher utilization rate of sample solution on the BAS Mem compared to the NC Mem. The finding verified the strong restriction for the flow direction of water on the BAS Mem, and indicated that the lateral-flow strips constructed with BAS Mem could enable more efficient flow of sample solution through the T and C lines, which helps to decrease the usage of sample solution required for detection. Consequently, the BAS Mem could be expected to drive more sample solutions carrying detectable complex to the T/C line for binding, so as to improve the detection signal on the T/C line.

Moreover, we analyzed the amounts of residual sample solution on membranes when BAS Mem and NC Mem were used to drive sample solution. We first constructed the mimic strips using two types of membranes, sample pads and absorption pads. Then we took the solution of nanogold-labeled antibody (Au-Ab) as the sample solution. We loaded 60 µL of sample solution onto the sample pad, recorded the flow of Au-Ab sample solution on the two types of membranes (Fig. 3F and Supplementary Movie 13, 14). The time-dependent variation of color intensity on membranes was shown in Fig. 3G, where the lower color intensity on the membranes signified the fewer residues of Au-Ab on the membranes. After 300 s, the color intensity on the BAS Mem was ~48% lower than that on the NC Mem (Fig. 3H), indicating a lesser residue of Au-Ab on the BAS Mem compared to the NC Mem. The above results provided evidence for the finding that the BAS Mem could reduce the residues of both nanogold-labeled antibodies and the "antigen-antibody-nanogold" complexes in the sample solution, thereby reducing background noise at non-detectable areas. As indicated previously, the application of BAS Mem could lead to a higher SNR of lateral-flow strips, so the BAS Mem could help to improve the sensitivity of LFA.

Finally, we visualized the advantages of using BAS Mem to construct the LFA strips, by plotting radar diagram. The radar diagram consisted of the characteristics, such as directionality and velocity of liquid flow, usage and residues of sample solution, and processing costs (Supplementary Fig. 15). We found the BAS Mem not only showed better performance in driving liquid, but also offered cost advantages in both processing and storage, compared to the NC Mem (Supplementary Notes 1)[31–33]. Therefore, the BAS Mem could be a potential component for LFA.

## Strips constructed with BAS Mem for ultra-fast hs-cTnI assay

We constructed the lateral-flow strips for the ultra-fast hs-cTnI assay using the BAS Mem, which was characterized by its fast, unidirectional flow of liquid and low residues of sample solution (Fig. 4). We first detected cTnI in sample solution by the lateral-flow strips (Fig. 4A and Supplementary Movie 15). We used PBS solution containing cTnI as the sample solution. ~1.1 s after the loading, the sample solution in the sample pad flowed to the conjugate pad and released the nanogold-labeled antibody from the conjugate pad. The nanogold-labeled antibody then bound with the cTnI antigen carried in the sample solution to form the detectable complex (referred to nanogold-labeled antibody-cTnI antigen). Thereafter, the sample solution carrying these complexes then flowed through the T line and C line on the BAS Mem, reaching the absorbent pad at ~32.6 s. Both the complexes and the nanogold-labeled antibody were selectively captured by the antibodies at T line and C line, respectively, to appear red color. Then the absorbent pad continuously absorbed the excess solution and completed the assay. The relationship between the color intensity at the T line and C line versus time was shown in Fig. 4B. The color intensity of the T line stabilized at 240 s, signifying the capability of lateral-flow strip to detect cTnI within 240 s.

We then evaluated the qualitative performance of the lateral-flow strips by detecting the sample solutions both with and without cTnI separately (Supplementary Movie 15, 16). When detected with the sample solution without cTnI, the lateral-flow strip showed significant red color only at the C line, corresponding to a singular peak in the color intensity graph below (Fig. 4C). In contrast, the lateral-flow strip detected with the sample solution containing cTnI exhibited colors at both the T and C lines, corresponding to two distinct peaks in color intensity graph below (Fig. 4D). Notably, both assays demonstrated minimal baseline noise. The results indicated that lateral-flow strips could qualitatively detect the inclusion of cTnI in the sample solution with a high SNR.

We also presented a comparison of the lateral-flow strips constructed with NC Mem and BAS Mem, in terms of detection time and sensitivity (referred to the SNR) in Supplementary Fig. 16. Compared to strips constructed with NC Mem, the lateral-flow strips constructed using BAS Mem could reduce the detection time by 73%. Additionally, the detection signal increased by 37.9%, and the background noise decreased by 67.7%. The SNR of lateral-flow strips constructed with BAS Mem was 19.3, which was 4.3 times higher than that of strips using NC Mem. Therefore, lateral-flow strips made of BAS Mem could reduce the detection time and enhance the detection sensitivity of LFA.

For the quantitative assessment, typically the LOD and the threshold of the assay, we prepared a series of sample solutions with varying concentrations of cTnI and detected them by lateral-flow strips (Fig. 4E). The color intensity at the T lines was plotted against the respective concentrations of cTnI (Fig. 4F), revealing a linear relationship (inset of the Fig. 4F). The LOD was 1.97 pg mL⁻¹. Using the sum of the average color intensity of the blank and three times its standard deviation (SD) as a benchmark, the threshold of the detection was 6.602 (Fig. 4G), as further supported by the Q-Q plot (Supplementary Fig. 17). These results indicated that the lateral-flow strips constructing with BAS Mem could achieve highly sensitive assay of cTnI with a low LOD.

Moreover, we further investigated the reliability of lateral-flow strips on clinical samples. We detected the serum sample of 25 suspected AMI patients using lateral-flow strips (Supplementary Fig. 18). The quantitative results suggested a specificity (true negative rate) of 100% (10/10) and a sensitivity (true positive rate) of 93.3% (14/15) for the hs-cTnI assays using the lateral-flow strips (Fig. 4H). The receiver operating characteristic (ROC) curve displayed an area under the curve (AUC) of 0.953, approaching the optimal AUC value of 1.0 (Fig. 4I), indicating that the nanogold lateral-flow based on BAS Mem could reliably identify the negative samples and the positive samples. And the cut-off value of the detection for the serum samples was 6.235, which was close to the threshold of the mimic sample results (6.602),

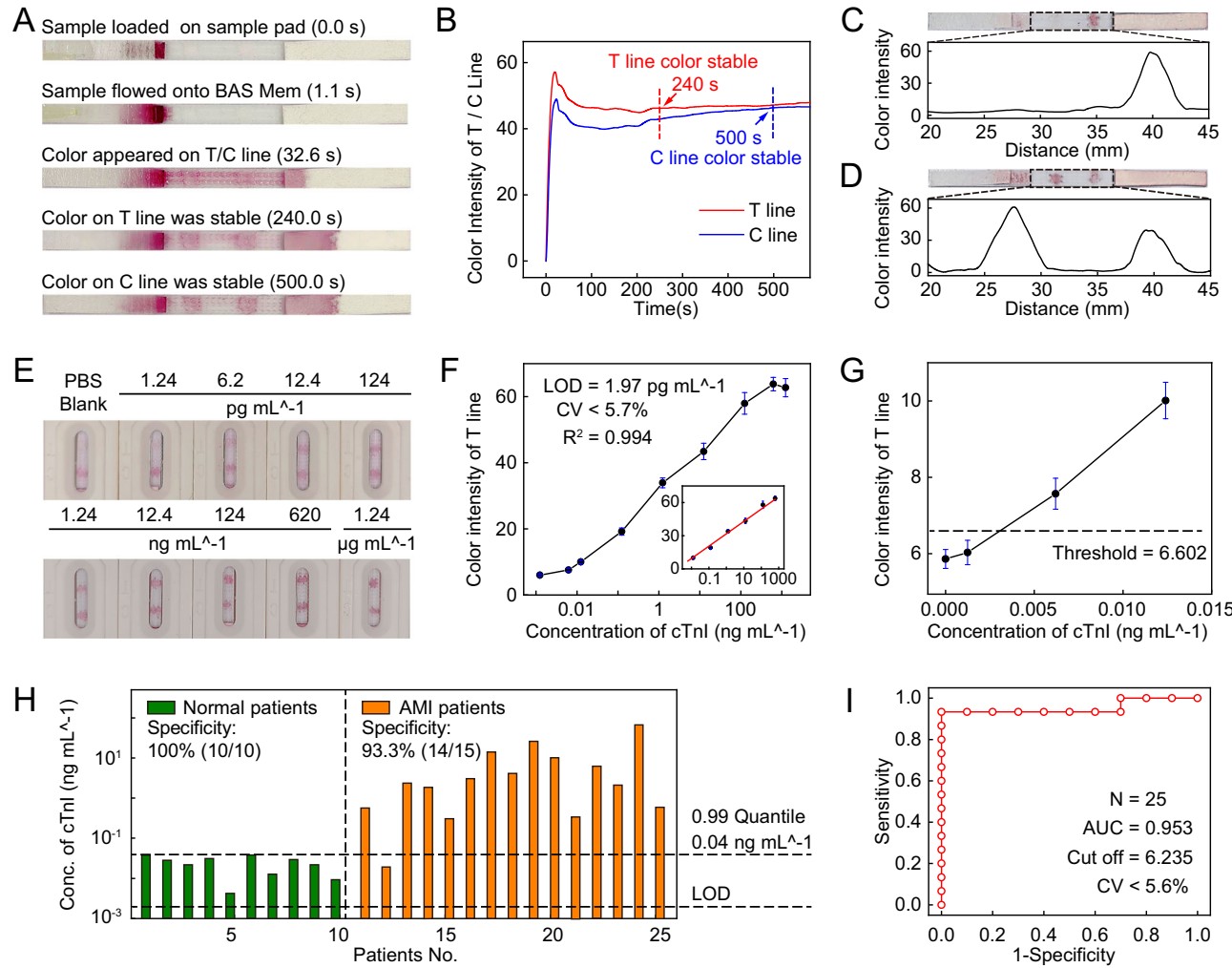

**Fig. 4 | The ultra-fast hs-cTnI assay using lateral-flow strips that constructed with BAS Mem. A** Optical images show the process of hs-cTnI assay. **B** Curves of color intensity at the T line and C line versus time. **C** The optical and graphical plots show the results of using lateral-flow strips to detect the sample solution without cTnI. Optical image shows the lateral-flow strip with color appeared at C line. The graph shows only a peak in color intensity at C line of the lateral-flow strip. **D** The optical and graphical plots show the results of using lateral-flow strips to detect the sample solution that with cTnI (124 ng mL⁻¹). Optical image shows the lateral-flow strip with color appearing at T and C line. The graph shows two peaks in color intensity at T line and C line of the lateral-flow strip. **E** Images show the lateral-flow strips that used to detect a batch of sample solution. **F** Plot shows the relationship between the concentrations of cTnI in the sample solution versus the color intensity on the T lines. **G** Plot shows the enlargement of detection results, when the concentrations of cTnI range from 0 pg mL⁻¹ to 12.4 pg mL⁻¹ in (**F**). **H** Histogram shows results of clinical serum samples tested by lateral-flow strips. Here, the green bars denote the quantitative results for regular patients using lateral-flow strips, while the red bars denote that for AMI patients. Here, the 0.99 quantile for cTnI concentration was 0.04 ng mL⁻¹. **I** ROC curves of the assays for clinical serum samples. Error bars represent SD in (**F**) and (**G**) $n = 3$, biologically independent replicates.

indicating that the highly sensitive assay was reproducible. Moreover, we displayed the detection of cTnI in fingertip blood using lateral-flow strips constructed with BAS Mem when AMI occurs, which provided the detection results within 4 min (Supplementary Movie 17).

**Comparison with other nanogold strips from well-known brands**
As mentioned above, the nanogold lateral-flow strips, incorporating the BAS Mem, offer an ultra-fast hs-cTnI assay. These lateral-flow strips delivered results in just 240 s, boasting a sensitivity of 93.3% and a specificity of 100%. The LOD of the ultra-fast hs-cTnI assay based on BAS Mem was 0.00197 ng mL⁻¹, which was comparable to the LOD of commercial-available hs-cTnI instruments[34–38]. We compared our lateral-flow strip with other commercial-available nanogold cTnI strips, including those commercially available from Tulip Group[39], Renesa[40,41], and Cortez[41], taking account of the detection time, sensitivity and specificity, LOD (Table 1). Notably, most commercial strips take between 15–30 min for the assay of cTnI, and the LOD is 3–5 orders of

magnitude greater than that of our work, even though their sensitivity and specificity are comparable.

## Discussion
We designed and prepared a BAS Mem facilitating ultra-fast unidirectional flow of liquid. The liquid on BAS Mem flowed forward unidirectionally and was restrained at the sharp corners, owing to the forces from the sidewalls of the structure units. The BAS Mem achieved a rectification coefficient of 14.5, which was the highest to our knowledge. For the unidirectional flow, the contact angle (θ) should lie between 48° and 64°. Moreover, the flow velocity of liquid could be regulated by changing the surface contact angle of the BAS Mem. We conducted a study on the mechanism of fluid control and proposed the principle of driving and pinning the liquid using the sidewalls of the BAS Mem. This mechanism of fluid control and principle can offer new ideas and methods for analyzing other fluid behaviors in the microfluidics field. Additionally, it may be widely applicable to solving

**Table 1 | Comparison showed the detection performance of our lateral-flow strips and commercial cTnI strips**

| Company / Source | Product | Method | Performance | | | |
|---|---|---|---|---|---|---|
| | | | Detection time | Sensitivity | Specificity | LOD (ng mL⁻¹) |
| Our work | | Nanogold LFA | ~ 4 min | 93.30% | 100% | 0.00197 |
| Tulip Group | Amicheck-Trop | | ~ 20 min | 85.70% | 92.80% | 0.1 ~ 0.3 |
| Renesa UG | CardioDetect | | 15 ~ 30 min | 86% | 93% | 7 |
| Cortez Diagnostics | RapiCard InstaTest | | 15 min | 98.90% | 99.10% | 1.5 |

technical problems related to fluid control. These characteristics made the BAS Mem promising for applications in microfluidics, droplet collection, and self-lubrication.

We used the BAS Mem as a lateral-flow membrane, which currently showed the highest rectification coefficient, to construct the fastest LFA that also featured high sensitivity. Firstly, the flow velocity of liquid was high on BAS Mem, but low at T line (comparable to that on NC Mem) (Supplementary Fig. 19). Then, the time was sufficient for the immunological binding between "antigen-antibody-nanogold" complex and antibody in the T line of our proposed lateral-flow strip (Supplementary Fig. 20, 21 and Supplementary Discussion 3). Building on these prerequisites, we increased the signal at T line and decreased the noise in non-detectable area by using BAS Mem. The SNR of lateral-flow strip was increased (Supplementary Fig. 16), so that enhanced the detection sensitivity of LFA.

Furthermore, the BAS Mem had excellent performance in driving sample solution unidirectionally when it was used as a component to construct the lateral-flow strip for ultra-fast hs-cTnI assay, as listed as follows. 1) The flow time of liquid on the BAS Mem ~93.5% lower than that on the NC Mem, indicating a shorter flow time of liquid on the BAS Mem compared to the NC Mem. 2) By restricting the flow direction of the liquid on the membrane, the BAS Mem enabled an efficient flow of sample solution, transporting a greater amount of detectable complex to the T line, so as to enhance the detection signal. 3) The residual sample solution on the BAS Mem was reduced by 48% compared to that in the NC Mem, minimizing background noise in non-detectable areas. Consequently, the BAS Mem in lateral-flow strips could be used to reduce the detection time and enhance the sensitivity of LFA. The results indicated the detection performance, including 1) a shortened detection time of 4 min, and 2) a lowered LOD to 1.97 pg mL⁻¹. The LFA based on the proposed lateral-flow strips achieved a specificity of 100% and a sensitivity of 93.3% in the detection of the serum samples of 25 suspected AMI patients. Compared to the performance of other commercial-available cTnI strips, the proposed lateral-flow strips saved 66.7%–86.7% of the detection time, and achieved a LOD that was lowered by 3–5 orders of magnitude with a low cost.

The BAS Mem, which transports liquid on the surface, offers a greater variety of sample solution types, a wider range of sizes for testable sample molecules and antibody-labeled signal amplification nanoprobes in LFA, compared to traditional fiber-based chromatographic membranes (such as NC Mem). This feature significantly broadens the applicability of the LFA, especially for those larger target biomarkers (such as certain virus particles or polysaccharides) and antibody-labeled signal amplification probe (such as gold particles and magnetic bead particles) that are challenging to process with traditional NC Mem. To verify this, we used the larger-diameter nanospheres (with a diameter of 500 nm) as the antibody-labeled signal amplification nanoprobe, and conducted the detection using two types of strips (Supplementary Fig. 22A). The strips made of NC Mem failed to complete the assay with higher background noise than the detection signal (Supplementary Fig. 22B). In contrast, the lateral-flow strips constructed with BAS Mem showed an increased signal at T line of 106.8%, and a decreased background noise of 79.0% compared to the strip constructed with NC Mem (Supplementary Fig. 22C). The SNR of lateral-flow strip made of BAS Mem was 11.7, which was 9.8 times

higher than that of the strip made of NC Mem (Supplementary Fig. 22D). The difference in the performance of SNR between the strips constructed with NC Mem and BAS Mem became more pronounced when larger-sized antibody-labeled signal amplification probes were used. We further utilized larger-sized microspheres (diameter 1 μm) as antibody-labeled signal amplification probes and applied them on the strips constructed with both NC Mem and BAS Mem. The results showed that the lateral-flow strips made with BAS Mem successfully completed qualitative detection, whereas those made with NC Mem did not (Supplementary Fig. 22E, F). The lateral-flow strips made with BAS Mem showed a 541.8% increase in the detection signal at the T line, and an 84.7% reduction in noise compared to the strips made with NC Mem (Supplementary Fig. 22G). The SNR of the lateral-flow strip was 19.2, which was 32 times higher than that of the strips made with NC Mem (Supplementary Fig. 22H). Therefore, the BAS Mem provides a broader selection space for target biomarkers and antibody-labeled signal amplification probes by facilitating fluid flow through its surface microchannels. The lateral-flow strip enables the detection of larger size of biomarkers and offers a solution to the retention of larger-sized antibody-labeled signal amplification probes that have been encountered recently.

Consequently, the BAS Mem demonstrated its excellent potential as a lateral-flow membrane to facilitate the ultra-fast hs-cTnI assay, which could be expected to realize an early warning of AMI and to further expand its applicability in public health emergencies, self-testing, food safety, and environmental analyzes.

## Methods
### Materials and reagents
PDMS (No. H052JCH032) was purchased from Dow Corning (Midland, Michigan, USA). Plastic membranes, including HDPE, PMMA, and PP, were commercially available. The cTnI antigen (No. CTNI-Ag5, >90%) was obtained from Fapon Biotech Inc. (Guangdong, China). The cTnI antibody (No. 700.485-3-5, ≥95%) and cTnI secondary antibody (No. H003, >95%) were purchased from OriGene Technologies, Inc. Co., Ltd (Wuxi, China). Human IgG (No. A065B202007027, >95%) was purchased from Yuangu Bio (Wuhan, China), rabbit anti-human IgG (No. KT165, >95%) was purchased from Biobomei biotech Co., Ltd (Hefei, China). Nanogold particles were prepared by the sodium citrate method. Gold (III) chloride solution (No. H1807013, 23.5~23.8%) was purchased from Aladdin (Shanghai, China). Acetone (No.2018110108, 99.5%) was purchased from Guangzhou Chemical Reagent Factory (Guangzhou, China). N, N-dimethylformamide (DMF, D112003, 99.9%) was purchased from Aladdin (Shanghai, China). 1-ethyl-3-(3-dimethyl-laminopropyl) carbodiimide hydrochloride (EDC.HCl, N808856, 98.5%) was purchased from Macklin (Shanghai, China). N-hydroxy succinimide (NHS, M01387, 98%) was purchased from Meryer (Shanghai, China). The fiberglass sample pads (No. SF-08) and conjugate pads (No. G-4), nitrocellulose chromatographic membrane (No. PALL90), neutral cotton absorption pad (No. H-5076), support backing pad (No. J-A6) and the plastic cases for the assay were obtained from Jingwen Technology (Shanghai, China). F-BSA (No. KT141, 98%) was purchased from Biobomei (Hefei, China).

The serum sample solutions were obtained from the Eighth Affiliated Hospital of Sun Yat-sen University. Our study has been

approved by the ethics committee (Medical Research Ethics Committee of the Eighth Affiliated Hospital of Sun Yat-sen University, 2023r113). All participants provided informed consent.

## Preparation of the BAS Mem

We designed with patterns of barbed arrow-like structure based on the design concepts shown in Supplementary Fig. 23 and Supplementary Methods 1, and used the techniques of laser carving, casting and hot embossing to successively prepare the BAS Mem. Firstly, we used a laser carving machining to carve the microchannels on the surface of an aluminum (Al) sheets to obtain an Al mold with same surface structure of BAS Mem. Then, we casted an ~1 mm-thick layer of PDMS prepolymer (the weight ratio of component A and B is 10:1) onto the Al mode. After degassing in a vacuum chamber, we heated the PDMS in an 80 °C oven for about 40 min. We peeled off the cured PDMS membrane to obtain the PDMS master. Subsequently, we sandwiched a HDPE membrane between the PDMS master and another flat glass slide to obtain a sandwich assembly. We placed this sandwich assembly on the platform of a hot compressor and embossed at 150 °C. After 5 min, we cooled down the temperature of the platform to room temperature. Finally, we removed the HDPE membrane from assembly and modified the surface of HDPE membrane with a hydrophilic coating according to the procedure outlined in Supplementary Fig. 24, and obtained the BAS Mem.

Here, we used SolidWorks 2021 (SOLIDWORKS, USA) software to design the patterns. We used the UV laser carving machine (UV-3S-SP, Dazu, China) to carve the microchannels on the Al sheet by laser, the scanning speed for laser carving was 400 mm s$^{-1}$. We used the DZF-6210 oven (AOKE Environmental Test, China) to heat the PDMS. The thickness of the HDPE membrane was 0.5 mm. We used the hot compressor (GS15011, Specac, UK) to thermoform the BAS Mem. The pressure applied to the assembly by the hot compressor was 0.3 tons. The composition of the hydrophilic coating, and methods for controlling its hydrophilicity were presented in Supplementary Methods 2.

## Characterizing the structure and the flow behavior of liquid on BAS Mem

After sputtering a thin layer of gold on the BAS Mem, we took the SEM images of the surface structure using a field emission scanning electron microscope (AURIGA, Zeiss, Germany). We used an inverted fluorescence microscope (Ti-U/B, Nikon, Japan) equipped with a high-speed video camera (Pco. edge 4.2 sCMOS, PCO, Germany) to take the videos of the unidirectional flow of the liquid on the BAS Mem.

## Construction of the lateral-flow strips

The procedures for constructing lateral-flow strips using the BAS Mem were as follows: 1) Prepared the T and C lines on the BAS Mem. 2) Coated the T and C lines of the BAS Mem with specific antigens/antibodies. 3) Prepared the components of the strips, including the sample pads, the conjugate pads, the BAS Mem with T and C lines, the absorbent pads, and the support backing pads. 4) Assembled all the components to form the lateral-flow strips.

We used cellulose acetate (CA) material to prepare the BAS Mem with T and C lines, taking into account the multiple advantages of CA in the preparation of lateral-flow strips (Supplementary Notes 2). We first added CA fiber bands onto the BAS Mem, which were used as the T and C lines of the lateral-flow strips. Firstly, we used the mixed solution of acetone and DMF to dissolve the CA powder and obtain a translucent slurry. We pipetted the 10 μL drop of slurry at the T or C line locations on the surface of the BAS Mem, and then added 5.0 μL of water to the slurry. The CA in the slurry could be rapidly transformed into the fiber bands. After washing the T/C line with water, we placed the BAS Mem with T and C lines on the hot plate at 37 °C, and immobilized the antibodies and antigens on the T and C lines using cross-linking agents, including EDC. HCl and NHS. We prepared the PBS solution (pH= 8.0)

containing 2.5% (w/w) EDC. HCl, and added it to the T and C lines of the BAS Mem at a dosage of 1 μL cm$^{-1}$. After 1 h, we prepared the PBS solution (pH = 8.0) containing 2.5% (w/w) NHS, and added it to equal volumes of cTnI antibody solution (1 mg mL$^{-1}$) and human IgG solution (1 mg mL$^{-1}$), respectively. Then we added the NHS-cTnI antibody mixture solution on the T line, and the NHS-human IgG mixture solution on the C line, respectively, at a dosage of 1.0 μL cm$^{-1}$. After another 1 h, we finished immobilization of antigens and antibodies on the T and C lines.

Then, we prepared other components that were used to construct the lateral-flow strips. We first cut the sample pads, conjugate pads, BAS Mem with T and C lines, absorbent pads, and support backing pads into standard sizes for assembling test strips. We then took 10 μL of antibody solution labeled with nanogold (50 μg mL$^{-1}$) and added it dropwise to the blank conjugate pad. The pad was placed on the 37 °C hot plate and incubated for 30 min away from light to obtain the conjugate pad.

Finally, we assembled all the components into a lateral-flow strip according to Supplementary Fig. 25A. We first tightly fixed the BAS Mem with T and C lines onto the center of the support backing pad. We then overlapped one end of the conjugate pad and absorbent pad on each side of the BAS Mem, respectively, and fixed the other end to the support backing pads. We overlapped one end of the sample pad on the conjugate pad and fixed the other end to the support backing pads as well. Here, the overlap length between each two components was ~2 mm. The assembled lateral-flow strip was then encapsulated within a white plastic casing. The lateral-flow cassette measures 75 mm in length and 20 mm in width. The center of the lateral-flow cassette features a hollowed-out area equipped with a sample hole, as well as designated test and control regions marked with "T" and "C", respectively (Supplementary Fig. 25B).

## The assays of lateral-flow strips

We loaded 60 μL of sample solution to the sample pad of the lateral-flow strip and read the result of assay after 4 min. There are three cases for results of assays: If both T and C lines show up, that's a valid positive result. If only the C line shows up, that's a valid negative result. If C line is invisible, that's an invalid result.

## Data acquisition and statistical analysis of immunoassay results

We took the optical images of lateral-flow strips which completed the immunoassay of cTnI under the white light, and used ImageJ software (National Institutes of Health, USA) to obtain the data, such as the color intensity of the T and C lines on the lateral-flow strips. We used Origin 2021 software (OriginLab, USA) to statistically analyze the data from the results, including plotting ROC assay curves, Q-Q plots, etc. The ROC curve and AUC values were used to analyze the cut-off value and accuracy of clinical serum sample measurement. The Q-Q plot was obtained with a confidence level of 99.7%, which was used to analyze the normal distribution of the results and the condition to calculate the threshold in the detection of mimic samples.

## Reporting summary

Further information on research design is available in the Nature Portfolio Reporting Summary linked to this article.

## Data availability

The data supporting the findings of the study are included in the main text and supplementary information files. Raw data can be obtained from the corresponding author upon request.

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

## Acknowledgements

This work was supported by the National Natural Science Foundation of China (No. 22174167), National Key R&D Program of China (2021YFA1400804), the Shenzhen Science and Technology Program (Grant No. JCYJ20190807160401657, JCYJ20220818102014028, JCYJ20230807111120043), and the Foundation of Guangdong Provincial Key Laboratory of Sensor Technology and Biomedical Instrument (No. 2020B1212060077). We would like to thank Fapon Biotech Inc. for providing reagent assistance.

## Author contributions

These authors contributed equally: J. H. Li, Y. R. Liu. J. H. Zhou, J. H. Li, and Y. R. Liu conceived the work and developed the methodology. J. H. Li designed the BAS Mem. J. H. Li and J. H. Du carried out the theoretical analysis. J. H. Li and T. Y. Wu fabricated the BAS Mem. J. H. Li, Y. R. Liu, C. P. Zhou, H. R. Liang, and T. Y. Wu performed the experiments. Z. H. Xiao performed preliminary experiments. J. H. Li and Y. R. Liu visualized data. J. H. Li and J. H. Zhou wrote the manuscript. Y. R. Liu, C. P. Zhou, and J. H. Zhou reviewed and edited the manuscript. All the writers commented on the paper. J. H. Zhou acquired funding and supervised the project.

## Competing interests

The authors declare no competing interests.
