## [Peer Review File · Nature Communications]

Barbed arrow-like structure membrane with ultra-high rectification coefficient enables ultra-fast, highly-sensitive lateral-flow assay of cTnIEditorial Note: Parts of this Peer Review File have been redacted as indicated to remove third-party material where no permission to publish could be obtained.

REVIEWER COMMENTS

Reviewer #1 (Remarks to the Author):

The manuscript proposed a barbed arrow-like structure membrane (BAS Mem) and demonstrated unidirectional and fast flow of liquid and the low residues of sample solution. The as-made strips could provide results within 240 s, with a limit of detection (LOD) of 1.97 pg/mL for cTnI. Basically, it is a kind of passive microfluidic design with structure features in sub millimeter or tens of micron scale and recently, there are some reports that have focused on the passive microfluidic design (Sci Rep 11, 21019 (2021); Sci Rep 11, 1986 (2021) and Microsyst Nanoeng 8, 62 (2022)). There are some issues to be discussed.

1. The authors claim that the flow is rectified by the barbed arrow-like structure. However, in LFA the flow starts from one end to the other and it is unidirectional. It could be seen as the flow path is cutted at the sample pad and there's no backflow path. Hence, rectification is actually not very meaningful in the regard.

2. The flow speed is increased which is clear and the LOD of 1.97 pg/mL is pretty good here. Usually, lower speed at the test line contribute to the increased sensitivity. But the authors have not address why this happens and what's the mechanism underlying. Another thing should taken into consider is the surface to volume ratio (SVR) of the substrate, which also contribute a lot to higher sensitivity. I don't think the SVR is good for the quasi-3D BAS in sub millimeter scale in comparison to the porous 3D structure of NC membrane. In this case, I think one point is very important. The authors should clarify why cellulose acetate (CA) is used to draw the T and C lines. What's the microscopic structure of the actual T or C line, for example by SEM? And what's the actual flow speed is at this kind of T line structure?

3. In order to obtain contact angles that meet the requirements, oxygen plasma treatment is selected to obtain different contact angles for further analysis. What is the duration of hydrophilicity. Usually it is not very stable and then this will affect the shelf time of the strip. Then, what kind of hydrophilic coating was chosen for use here?

In general, I do not think the novelty and significance of the manuscript meet the criteria of the journal.

Reviewer #2 (Remarks to the Author):

Dear Author,

This manuscript focused attention on the development of novel lateral-flow assay to improve its detecting speed and efficiency. Use of unidirectional liquid transport mechanism on disease diagnosis is very interesting. The following points should be reconsidered.

(1) Barbed arrow-like groove is designed to realize the unidirectional liquid transport. The test experiments also validate that this groove can enhance transport speed and reduce the residue of sample solution. However, all these results are depended on the parameter of groove structure. It is

very important to show how to determine and how to design.

(2) Oxygen plasma for different periods of time was used to adjust the surface wettability. This surface treatment can not remain long period. In practical application, how do you adjust surface wettability.

(3) It will much more better to define unidirectional liquid transport coefficient than just equation derivation as like Eq. (1)-(2).

(4) This manuscript shows one practical application in disease diagnosis. All the line or parameter setting should consider the test demand or practical application. For example, how to determine the T line and C line in Fig.3.

(5) The organization of this manuscript should be polished.

Replies to reviewers' comments

To Reviewer #1:

Referee letter: *The manuscript proposed a barbed arrow-like structure membrane (BAS Mem) and demonstrate unidirectional and fast flow of liquid and the low residues of sample solution. The as-made strips could provide results within 240 s, with a limit of detection (LOD) of 1.97 pg/mL for cTnI. Basicly, it is a kind of passive microfluidic design with structure features in sub millimeter or tens of micron scale and recently, there are some reports that have focused on the passive microfluidic design (Sci Rep 11, 21019 (2021); Sci Rep 11, 1986 (2021) and Microsyst Nanoeng 8, 62 (2022)). There are some issues to be discussed.*

Response: Thank you for the comment. As the reviewer pointed out, the barbed arrow-like structure membrane (BAS Mem) we proposed is a type of passive microfluidic design. The methods of fluid control in microfluidics are divided into two types: the active microfluidics and the passive microfluidics. The active microfluidics often require the input of external energy (such as applying an electric field or pressure) to drive and control the flow of fluid in microchannels. The passive microfluidics, on the other hand, does not require external energy input but relies on the properties of the fluid itself, and the design of the microchannels to control the fluid. For example, capillary forces in microchannels can be used to drive the flow of fluid, or the structure of microchannels can be designed to control the flow direction of the fluid, using the differences in resistance of fluid within the channels. Due to its independence from energy input and its ease of integration and operation, the passive microfluidics is well-suited for developing portable, cost-effective, and rapid diagnostic devices for on-site testing. Thus, the passive microfluidics is of significance in microfluidic technologies.

1) In our work, we proposed a BAS Mem that enabled the unidirectional, rapid, and low-residual flow of liquid. The BAS Mem uses the sharp edges on the sidewalls of microchannels to prevent the wetting and spreading of liquid, and making use of the surface tension of liquid and capillary forces to drive the liquid, thereby achieving the unidirectional transport of liquid. The BAS Mem exhibits excellent rectification capabilities, **performing the highest rectification coefficient that reported to date**. We conducted a study on the mechanism of fluid control, and proposed the principle of driving and pinning the liquid using the sidewalls of the BAS Mem. This mechanism of fluid control can offer new ideas and methods for analyzing other flow behaviors in the microfluidics. Additionally, it may be widely applicable to solving technical problems related to fluid control. As an example of application, we used the BAS Mem to prepare batches of lateral-flow strips, and **developed a fastest lateral-flow assay (LFA) for highly sensitive cTnI (hs-cTnI; cTnI is a typical cardiac biomarker) assay**. The experimental results indicated that, employing the BAS Mem in lateral-flow strips could reduce the detection time and improve the signal to noise ratio (SNR) of LFA. This strategy of constructing lateral-flow strips with the BAS Mem provides a new and efficient method for the field of point-of-care testing (POCT).

2) We listed the several passive microfluidic designs mentioned by the reviewer (Figure R1), and compared them with our design (Table R1). Lin et al. (Sci Rep 11, 21019 (2021)) introduced a passive microfluidic device for the loop-mediated isothermal amplification (MF-LAMP). The MF-LAMP chip contained 3 channels each with 6 concentric circular chambers in a row. The concentric circular chamber contained a reaction chamber, a capillary channel, and a peripheral channel. The width of

the capillary channel directly affected the solution substitution for fluid injection, and oil enclosure efficiency in this system (Figure R1A). MF-LAMP successfully reduced the liquid loss and condensation, enhancing the reliability and efficiency of gene testing and expression analysis in the POCT field. Uddin et al. (Sci Rep 11, 1986 (2021)) proposed a microfluidic lab on chip (LOC) device for conducting personalized 96-well ELISA for individual patients. In terms of the fluid control, they used the polydimethylsiloxane (PDMS) with microcavities as the main body for pumps and valves, and a roller rod pressed against the PDMS to toggle the pumps and valves (Figure R1B). The key to driving the liquid in the device was creating a negative pressure in the microchannels, and the negative pressure drove the liquid into the reaction chamber by the pressure differences on two sides of the liquid surfaces. This strategy of fluid control could not only precisely control the flow velocity, but also address the issue of inconsistencies in reaction times between different steps of the assay. Tang et al. (Microsyst Nanoeng 8, 62 (2022)) summarized the geometric principles of passive and label-free microfluidics for the separation of biological micro-objects in this review, primarily including the shape modification, 3D structures, topology modification and their combinations. They also compiled different schemes for passive label-free microfluidics, such as deterministic lateral displacement (DLD), pinched flow fractionation (PFF), viscoelastic microfluidics (VEM) and inertial microfluidics (IMF) (Figure R1C).

[REDACTED]

Figure R1. Various designs of passive microfluidics for flow control. A) Schematic illustration of the passive-driven MF-LAMP chip (reproduced from Sci Rep 11, 21019 (2021)). B) Conceptual illustration of the 96-well associated LOC platform (reproduced from Sci Rep 11, 1986 (2021)). C) Geometric innovations in microfluidic schemes which enabled the separation of particle through structural design (reproduced from Microsyst Nanoeng 8, 62 (2022)).

3) Unlike the passive microfluidics design mentioned in Table R1 above, we proposed a barbed

arrow-like structure on a plastic membrane and tuned its surface hydrophilicity to achieve the precise control of liquid flow (Figure R2). We found that sharp edges on the sidewalls of microstructures could provide flow resistance to fluid, preventing the liquid from backward wetting on the BAS Mem. Meanwhile, the liquid could be driven forward by its surface tension and the capillary forces, so that the BAS Mem can achieve the unidirectional transport of fluid. This BAS Mem shows the highest rectification coefficient known to date. The unidirectional and fast flow of liquid, and the low residues of sample solution on BAS Mem were illustrated in experiments. Furthermore, we have developed an ultra-fast, high sensitivity LFA assay for cTnI using the lateral-flow strips made of BAS Mem. This lateral-flow strip provided results in just 4 min, and was highly suitable for the use in emergency and resource-limited settings. It aided in the rapid diagnosis of typical acute myocardial infarction (AMI) symptoms, such as chest pain, and provided early warning of the onset of AMI. The lateral-flow strip we proposed not only meets the current needs for home-health medicine and on-site rapid detection, but also brings some new ideas to the field of medical diagnostics.

Figure R2. The illustration of the ultra-fast hs-cTnI assay for early warning of AMI, by using a lateral-flow strip constructed with BAS Mem that drives the flow of liquid unidirectionally by its side walls.

To make this point clearer, we have added statements to describe the type of fluid control of the fiber-based chromatographic membranes in the **Introduction** section of the revised manuscript. “In this method, it is evidently crucial for lateral-flow membrane to drive the flow of the sample solutions through passive microfluidics, and to selectively capture the complexes carried in the solution¹⁴.” (See page 2, line 36-38)

Also, in order to sharpen the novelty of this work and demonstrate its broad implications, we have revised the manuscript accordingly.

(1) We have revised statements on theoretical part of this work in the **Discussion** section of revised manuscript. “We conducted a study on the mechanism of fluid control, and proposed the principle of driving and pinning the liquid using the sidewalls of the BAS Mem. This mechanism of fluid control can offer new ideas and methods for analyzing other fluid behaviors in the microfluidics field. Additionally, it may be widely applicable to solving technical problems related to fluid control.” (See page 18, line 355-358)

(2) We have added statements on the degree of advances in the application of BAS Mem in LFA in the **Discussion** section of revised manuscript. “We used the BAS Mem as a lateral-flow membrane, which currently showed the highest rectification coefficient, to construct the fastest LFA that also featured high sensitivity.” (See page 18, line 360-361)

(3) We have also added the statements and experimental demonstration to show the impact of this work in other research areas in our response letter. We presented an example application of using

capillary tubes prepared with BAS Mem for passive sampling (see Figure R16). This demonstrates that the BAS Mem can also be developed as the passive unidirectional flow pipette to avoid contamination caused by the back-and-forth flow of samples in medical testing. Therefore, our work not only provides a new method for medical testing, but also has potential applications in multiple fields including materials chemistry, surface science, fluid dynamics, and public health. (See page 23 of the response letter)

Comment 1: *The authors claim that the flow is rectified by the barded arrow-like structure. However, in LFA the flow starts from one end to the other and it is unidirectional. It could be seen as the flow path is cutted at the sample pad and there's no backflow path. Hence, rectification is actually not very meaningful in the regard.*

Response: Thank you for your comment. In traditional sandwich method of nanogold LFA, once the sample solution was applied to the sample pad of the strip, the solution would flow from one end to the other along the predefined path, owing to the capillary force of chromatographic membranes. Basically, as shown in Figure R3, the sample solution would first immerse the conjugate pad after being added to the LFA strip. The antigen in sample solution could bind with gold-labeled secondary antibodies in conjugate pad, and formed “antigen-antibody-nanogold” complexes. The complexes would then flow into the nitrocellulose membrane (NC Mem, which was the most widely used chromatographic membrane at present), and reached the test (T) and control (C) lines on the membrane, where they could bind to the specific antibodies and develop color. Subsequently, the solution would enter the absorbent pad under the capillary action of the NC Mem, and completed the assay. During this process, part of the sample solution flowed from the conjugate pad, flowed along the NC Mem and flowed into the absorbent pad.

However, it was notable that: (1) The flow of sample solution relied on the capillary force of NC Mem. Usually, the flow of liquid from the conjugate pad to the T line (distance L in the figure) usually took 0.5-1 min. This time was incorporated into the overall detection time for LFA. The slower the flow velocity in this process, the longer the detection time required. (2) Although the sample solution flowed past the T and C lines towards the absorbent pad, there was still a part of sample solution that remained in the front of the T and C lines of the NC Mem. In other words, a fraction of the sample solution failed to reach the T and C lines, and the containing “antigen-antibody-nanogold” complexes did not bind with the antibodies in T line to form detection signal (which meant that the sample solution was not fully utilized), leading to the decrease of the detection signal of LFA. (3) The residual sample solution remained in the NC Mem (in point (2)) would increase the background noise, resulting in the decrease of SNR (which was proportional to the detection sensitivity). Therefore, as shown in Figure 3B, the strategy for improving the performance of LFA could be listed as: 1) Increasing the flow velocity of the liquid in the non-detection areas to reduce the detection time of LFA. 2) Directing more sample solution flowed through the T and C lines, and the containing “antigen-antibody-nanogold” complexes were captured to form the detection signal, so that increased the detection signal. 3) Reducing the residual of sample solution on the membrane to decrease the background noise. These improvements can increase the SNR of LFA, thereby enhancing the sensitivity of the LFA.

Figure R3. The schematics showing the flow path of liquid during the detection process of a lateral-flow assay, and the strategies to improve the performance of LFA. A) The schematic illustration shows a typical nanogold lateral-flow assay strip based on the nitrocellulose membrane (NC Mem). The graph details the structure of the strip, the liquid flow path, and the process of signal generation and detection at the T (test) line and the C (control) line. The area marked by dotted boxes specifically indicates the source of the signal and noise of LFA. B) Strategies to reduce the detection time, and enhance the detection sensitivity of LFA.

In response to the issues discussed above, we proposed a barded arrow-like structure membrane to rectify the flow of liquid. We used membranes with this structure (BAS Mem) to replace the traditional NC Mem, and constructed a novel lateral-flow strip. The BAS Mem enabled the unidirectional flow of liquid on the surface. The rectification of the liquid flow on the surface of BAS Mem provided three advantages for LFA as follow:

1) The unidirectional transport of BAS Mem for the liquid flow helped to increase the flow velocity of the liquid on the BAS Mem, so that reducing the flow time from the conjugate pad to the T line, so that shortened the detection time for LFA.

We applied 10 μL of red ink to the surface of the BAS Mem and the NC Mem respectively, and observed the flow behavior of the liquid. As shown in Figure R4A, the red ink on the BAS Mem could spread unidirectionally forward, while the red ink flowed both forwards and backwards on the NC Mem. We also compared the flow velocity of the liquid on the two types of membranes, as shown in Figure R4B. The average flow velocity was 0.32 mm/s on NC Mem. In contrast, the average flow velocity reached 5.32 mm/s on the BAS Mem. The flow time of the liquid on the BAS Mem was reduced by 93.5% compared to that in NC Mem (Figure 4C). The results suggested that the unidirectional rectification of BAS Mem for the liquid helps to shorten the detection time for LFA.

Figure R4. Characterization of flow velocity for the unidirectional flow of liquid on BAS Mem, and the flow of liquid on NC Mem. A) Optical images showing the time-dependent flow of liquid on two types of membranes. Scale bar = 5 mm. B) Histogram showing the average velocities of liquid flowing on the membranes. C) Histogram showing the time of liquid flowing on the membranes.

2) The unidirectional transport of BAS Mem for the liquid flow helped to increase the detection

signal for the same volume of sample solution, thereby enhancing the detection sensitivity.

We compared the lateral-flow strips made of NC Mem and BAS Mem in detecting the sample solutions of the same volume and concentration of cTnI (1.24 ng/mL). Figure R5 showed the color development at the T and C lines of two types of strips with the same antibody-labeled signal amplification nanoprobe, as well as the detection signal at the T line. For the use of gold particles (with a diameter of 30 nm) as the antibody-labeled signal amplification nanoprobe, the strips constructed with the two membranes displayed clear color at both the T and C lines (Figure 5A). The lateral-flow strip constructed with BAS Mem showed a higher signal at the T line (Figure 5B), exhibiting an increase of 37.9% compared to the strip constructed with NC Mem (Figure 5C). Furthermore, we used the larger-diameter nanospheres (with a diameter of 500 nm) as the antibody-labeled signal amplification nanoprobe, and conducted the detection using two types of strips. It was difficult to clearly distinguish the T and C lines in the strips constructed with NC Mem (Figure R5D). In contrast, the lateral-flow strips constructed with BAS Mem still displayed clear color on the T and C lines, with a significantly increased signal at the T line (Figure R5E), showing an increase of 106.8% compared to that in the strip constructed with NC Mem (Figure R5F). The results indicated that the BAS Mem could transport more sample solution and antibody-labeled signal amplification nanoprobe to reach and flow through the T and C lines (even the antibody-labeled signal amplification nanoprobe with a diameter of up to 500 nm would not remain on the BAS Mem as well), thereby achieving higher signal with the same volume of sample solution.

Antibody-labeled signal amplification nanoprobe:

30 nm-gold particle

Antibody-labeled signal amplification nanoprobe:

500 nm-nanosphere

Figure R5. Results of lateral-flow strips constructed with NC Mem and BAS Mem for the

assay of positive sample solution using different types of antibody-labeled signal amplification nanoprobe. A-C) Using 30 nm-gold particle as the antibody-labeled signal amplification nanoprobe. D-F) Using 500 nm-nanosphere as the antibody-labeled signal amplification nanoprobe. A) and D) Optical images showing the lateral-flow strips that constructed with NC Mem and BAS Mem, respectively, with visible color on membranes. Scale bar = 5 mm. B) and E) The plots showing the variation of color intensity on lateral-flow strips constructed with NC Mem and BAS Mem, respectively. C) and F) Histograms showing the signal at T line increased using lateral-flow strip constructed by BAS Mem, compared to NC Mem.

Therefore, the characteristic that the BAS Mem unidirectionally transport the sample solution, enabled more sample solution to flow toward and flow through the T and C lines, and increased the signal with the same volume of sample solution, so that helping to enhance the sensitivity of LFA detection. Specifically, it was worth mentioning that using BAS Mem for the liquid transport could reduce the residual of sample solution on the membrane, thereby decreasing the sample volume required for detection. As shown in Figures 3D-E in the manuscript, the optical images showed different volumes of liquid flowing on the BAS Mem. It could be seen that even a small amount of sample solution could flow completely to the end on the BAS Mem. This result indicated that it is possible to use less sample solution (such as finger-prick blood) to complete the detection without sacrificing detection sensitivity and accuracy in practical applications. Therefore, the use of BAS Mem in LFA not only lowers testing costs, but also makes the lateral-flow strip more suitable for resource-limited situations or when conserving samples is necessary.

3) The unidirectional transport of BAS Mem for the liquid flow also helped to reduce the background noise in LFA, thereby enhancing the sensitivity of LFA.

The traditional fiber-based chromatographic membranes (such as NC Mem) are typically made from fiber filaments. These materials of chromatographic membranes will absorb the sample solution on one hand. On the other hand, the materials will also non-specifically adsorb protein molecules (such as antigen) in the sample solution due to their characteristic of hydrophobic. This might lead to the residue of protein molecules in non-detection areas, and increasing the noise of detection in LFA. Additionally, the pore size of the fiber-based chromatographic membranes is usually very small. When the sample solution contains large size of protein molecules or antibody-labeled signal-amplifying nanoprobe, these protein molecules and nanoparticles are often difficult to pass through the tiny pores of the fiber membrane, and are easily retained in the membrane. This might also lead to the residue of antibody-labeled signal-amplifying nanoprobe in non-detection areas, and increasing the noise of detection in LFA. When there is significant background noise in the non-detection areas of the lateral-flow strip, it will greatly increase the SNR of LFA. Even the background noise will exceed the actual detection signal, and preventing the lateral-flow strip from detecting low-concentration sample solutions. In contrast, BAS Mem is made by using the plastics as the substrate materials, such as high-density polyethylene (HDPE) and polymethyl methacrylate (PMMA). The liquid can flow unidirectionally under the driving force generated by the sidewalls of the microchannels of the BAS Mem. This material and structural design not only effectively reduces the residual of sample solution on the membrane (as described in (2) above), but also decreases the non-specific adsorption of protein molecules and the residue of antibody-labeled signal amplification

nanoprobes in the membrane.

We first experimentally compared the non-specific adsorption of protein molecules on NC Mem and BAS Mem, considering the non-specific adsorption of protein molecules is one of the factors affecting the background noise of detection in LFA. We used the NC Mem and the BAS Mem (BAS Mem was made of PMMA substrate) to transport solutions of fluorescein-labeled bovine serum albumin (F-BSA) and compared the residual non-specific adsorption of F-BSA on both types of membranes. As shown in Figure R6A, the fluorescence intensity on the surface of the BAS Mem was lower after the F-BSA solution flowed through the BAS Mem and NC Mem, indicating less residual F-BSA solution on the BAS Mem. Then we flushed both membranes with phosphate buffer solution (PBS) to remove the F-BSA solution that hadn't run out. The results showed that the BAS Mem still exhibited lower fluorescence intensity after PBS flushing (Figure R6B). This result suggests that the HDPE material of the BAS Mem is less likely to non-specifically adsorb protein molecules from the sample solution when BAS Mem is used to transport the sample solution, thereby the use of BAS Mem helping to reduce the background noise in LFA.

We also compared the non-specific adsorption of F-BSA solution on BAS Mem, well plates, and micro-reaction chambers to further demonstrate the impact of flow behaviors of liquid on non-specific adsorption.

We used PMMA substrate to fabricate the BAS Mem, and then observed the non-specific adsorption of protein molecules on the material surface, under three classic flow behaviors of liquid: unidirectional flow, shaking, and standing. (These flow behaviors are typically used in LFA, enzyme-linked immunosorbent assay and chemiluminescence assay, respectively.) We made the F-BSA solution flow unidirectionally on the BAS Mem, placed the solution on a shaking BAS Mem, and left the solution on standing the BAS Mem. After 240 s in each condition, we used PBS to flush the surface of each BAS Mem. The fluorescence microscopy images of each BAS Mem were shown in Figure R6C. For the condition that liquid flowed unidirectionally on the BAS Mem, the fluorescence intensity of F-BSA on the BAS Mem was the lowest, indicating the least residual of F-BSA on the membrane. Furthermore, after flushing the BAS Mem with PBS, the fluorescence intensity of F-BSA on the BAS Mem was also the lowest, suggesting the lowest degree of non-specific adsorption of F-BSA protein molecules on the BAS surface. In contrast, the fluorescence intensity was higher in the shaking and standing tests, indicating more non-specific adsorption of protein molecules (Figure R6D). The results demonstrate that unidirectional flow of the sample solution on the BAS Mem can help reduce non-specific adsorption of protein molecules.

Therefore, the plastic materials used to prepare the BAS Mem are less likely to non-specifically adsorb sample molecules in the proposed LFA assay. Additionally, the unidirectional flow of liquid on the BAS Mem also helps to reduce the non-specific adsorption of protein molecules. In short, using BAS, which can transport sample solutions unidirectionally, helps to minimize the non-specific adsorption of protein molecules in non-detection areas, thereby reducing background noise.

Figure R6. Characterization of residues of F-BSA on two types of membranes. A) Optical images of the membranes after transporting F-BSA and flushing by PBS. The BAS Mem was fabricated with HDPE substrate. Scale bar = 0.5 mm. B) Histogram showing the fluorescence intensity of F-BSA on NC Mem and BAS Mem after F-BSA flowing and PBS flushing. C) Optical images of the membranes after transporting F-BSA and flushing by PBS, when the flow behaviors of liquid on BAS Mem are unidirectional flow, shaking and standing. The BAS Mem was fabricated with PMMA substrate. Scale bar = 0.5 mm. D) Histogram showing the residual fluorescence intensity of F-BSA on BAS Mem, when the flow behaviors of liquid on BAS Mem are unidirectional flow, shaking and standing.

Furthermore, we prepared lateral-flow strips using both NC Mem and BAS Mem, and experimentally compared the retention of antibody-labeled signal amplification nanoprobe in non-detection areas. Based on the results of Figure R5, Figure R7 additionally displays the noise in the non-detection areas and the SNR of two types of strips. When the antibody-labeled signal amplification nanoprobe was gold particles (30 nm in diameter), both strips completed the detection (Figure R7A-B). But, the lateral-flow strip constructed with BAS Mem exhibited lower background noise (Figure R7C), showing a 67.7% decrease compared to the strip made with NC Mem. The SNR for the lateral-flow strip made of BAS Mem was 19.3, which was 4.3 times higher than that of the strip made of NC Mem. Additionally, for the LFA using nanospheres (500 nm in diameter) as antibody-labeled signal amplification nanoprobe, the strip constructed with NC Mem failed to complete the detection (because the background noise was higher than the actual detection signal) (Figure R7A). In contrast, the lateral-flow strip made of BAS Mem still managed to perform the detection, showing an 79.0% decreased background noise compared to that in the strip made of NC Mem. The SNR lateral-flow strip made of BAS Mem was 11.7, which was 9.8 times higher than that of the strip made of NC Mem (Figure R7H). The results indicated that lateral-flow strip constructed with BAS Mem effectively reduce the retention of antibody-labeled signal amplification nanoprobe (such as large-diameter nanospheres), thereby decreasing the detection background noise. Therefore, the BAS Mem could reduce the non-specific adsorption of protein molecules and the retention of antibody-labeled signal amplification nanoprobe on the BAS Mem, which helps to reducing the

background noise in LFA.

Figure R7. Results of lateral-flow strips constructed with NC Mem and BAS Mem for the assay of positive sample solution using different antibody-labeled signal amplification nanoprobe. A-D) Using 30 nm-gold particle as the antibody-labeled signal amplification nanoprobe. E-H) Using 500 nm-nanosphere as the antibody-labeled signal amplification nanoprobe. A) and E) Optical images showing the lateral-flow strips that constructed with NC Mem and BAS Mem, respectively, with visible color on membranes. Scale bar = 5 mm. B) and F) The plots showing the variation of color intensity on the lateral-flow strips constructed with NC Mem and BAS Mem, respectively. C) and G) Histograms showing the increase of signal and the decrease of noise using lateral-flow strips constructed by BAS Mem, compared to NC Mem. D) and H) Histograms showing the SNR increased using lateral-flow strips constructed by BAS Mem, compared to NC Mem.

Therefore, as shown by the results in Figures R5, R6, and R7, the characteristic of BAS Mem to transport sample solutions unidirectionally helps to enhance the signal at the T line, and reduce background noise in the non-detection areas, and thereby improve the SNR of LFA.

In summary, the ability of BAS Mem to transport liquid unidirectionally not only helps to shorten the detection time of LFA, but also increases the detection signal and decreases background noise, thus enhancing the sensitivity of LFA. Therefore, the rectification of BAS Mem for flow of liquid is meaningful.

Also, in order to illustrate the rectification effect of the BAS Mem on liquid, and the advantages of using BAS Mem in LFA, we have revised the manuscript accordingly as follow.

(1) To illustrate that the rectification of liquid (that offered by BAS Mem) could help to reduce the flow time of the liquid on membranes, and contributed to the reduction of detection time in LFA, we have revised the descriptions that liquid on BAS Mem has a shorter flow time compared to NC Mem in the **Results** section of the revised manuscript. **“The flow time of liquid on the BAS Mem**

approximately 93.5% lower than that in the NC Mem (Fig.3C), indicating a shorter flow time of liquid on the BAS Mem compared to the NC Mem. Further calculations revealed that the average flow velocity on the BAS Mem was 5.48 mm/s, which was approximately 23 times faster than in the NC Mem (0.23 mm/s) (Supplementary Fig. 14). These results suggested the BAS Mem enabled shorter flow time of liquid, comparing to the NC Mem. Consequently, the BAS Mem has the potential to shorten the time of sample solution taken to flow to the T and C lines on lateral-flow strips, and has contributed to the reduction of detection time in LFA.” (See page 11, line 226-232)

In order to show the lateral-flow strips made of BAS Mem have shorter detection time than the classic LFA strips clearly, we have added the comparison statement and graph of the detection time for using strips made of NC Mem and lateral-flow strips made of BAS Mem. “We also presented a comparison of the lateral-flow strips constructed with NC Mem and BAS Mem, in terms of detection time and sensitivity (referred to the SNR) in Supplementary Fig.16. Compared to strips constructed with NC Mem, the lateral-flow strips constructed using BAS Mem could reduce the detection time by 73%.” (See page 15, line 300-302)

(2) To illustrate that the rectification of liquid (that offered by BAS Mem) could help to reduce the usage of sample solution, and contributed to the increase the utilization rate of the sample solution in LFA, we have added the comparison statement and graph in the **Results** section of the revised manuscript. “The usage of sample on the BAS Mem approximately 70% lower than that in the NC Mem (Fig.3E), indicating a higher utilization rate of sample solution on the BAS Mem compared to the NC Mem.” (See page 12, line 239-241)

(3) To illustrate that the rectification of liquid (that offered by BAS Mem) could help to increase the signal at the T line and decrease the noise in non-detection areas, and thereby enhance the detection sensitivity (by improving the SNR) in LFA, we have added a comparative detection by using the strips prepared with NC Mem and BAS Mem in the **Results** section of the revised manuscript. “We also presented a comparison of the lateral-flow strips constructed with NC Mem and BAS Mem, in terms of detection time and sensitivity (referred to the SNR) in Supplementary Fig.16.” (See page 15, line 300-301) “Additionally, the detection signal increased by 37.9%, and the background noise decreased by 67.7%. The SNR of lateral-flow strips constructed with BAS Mem was 19.3, which was 4.3 times higher than that of strips using NC Mem. Therefore, lateral-flow strips made of BAS Mem could reduce the detection time, and enhance detection sensitivity of LFA.” (See page 15, line 302-305)

Comment 2: The flow velocity is increased which is clear and the LOD of 1.97 pg/mL is pretty good here. Usually, lower velocity at the test line contribute to the increased sensitivity. But the authors have not address why this happens and what’s the mechanism underlying.

Response: Thank you for the comment.

1) We agree with the reviewer’s point that usually, lower velocity at the test line contribute to the increased sensitivity. Taking the classic sandwich method nanogold LFA strip as an example, slowing down the flow velocity of the sample solution at the T line could prolong the contact time between the “antigen-antibody-nanogold” complexes and the antibodies immobilized at the T line. This extended time can improve the efficiency of immunological binding between “antigen-antibody-nanogold” complexes and the antibodies, thereby increasing the detection signal (increased color intensity) at the T line and enhancing the sensitivity of LFA.

2) In our work, the reasons why our proposed lateral-flow strip could transport liquid at an increased velocity, while maintaining a low limitation of detection (LOD), were that we had met the following two key prerequisites. Prerequisite 1: The flow velocity of liquid was high on BAS Mem, but low at T line (Figure R8A). Prerequisite 2: Time was sufficient for the immunological binding between “antigen-antibody-nanogold” complex and antibody immobilized at the T line (Figure R8B). By meeting these two prerequisites, the BAS Mem that enabled the unidirectional flow of liquid, was able to (as we pointed out in **comment 1**): increased the utilization of the sample solution, so that increasing the detection signal; decreased the residue of the sample solution, the non-specific adsorption of target protein molecules, and the retention of antibody-labeled signal amplification nanoprobes, so that decreasing the background noise; therefore, increased the SNR of lateral-flow strip, which in turn enhanced the sensitivity of LFA (Figure R8C).

Figure R8. Conceptual diagram showing the strategies for increasing flow velocity of liquid, and enhancing the detection sensitivity of lateral-flow strips. A) Schematic diagram showing the prerequisite 1, that the flow velocity is high on BAS Mem, but low at T line. B) Schematic diagram showing the prerequisite 2, that time is sufficient for the immunological binding between “antigen-antibody-nanogold” complex and antibody. C) Schematic diagram showing the increase of signal and the decrease of noise, and the improvement of the SNR when using the lateral-flow strip constructed by BAS Mem, compared to NC Mem.

Meeting prerequisite 1: We only increased the flow velocity of the sample solution in the non-detection areas (which referred to the BAS Mem) of the lateral-flow strip, without accelerating the flow velocity of solution in the detection areas (which referred to T and C lines) in this work.

We first supplemented experiment to test the flow velocity of the liquid in the T and C lines. We used the PMMA substrate to fabricate the BAS Mem and prepared the T and C lines on the BAS Mem. We employed F-BSA solution as a tracer, and observed the flow behavior of liquid in the lines. After the loading of F-BSA solution, it continuously penetrated into the T line over time. Figure R9A showed the noticeable change in fluorescence intensity with time. At approximately 8.0 s, the fluorescence intensity reached stable (Figure R12B), with no significant changes occurring thereafter. Therefore, we could calculate the average flow velocity of the liquid in the T line, which was about 0.25 mm/s. The speed was similar to that on NC Mem, which is 0.32 mm/s (Figure R4).

Figure R9. Characterization of flow velocity of F-BSA solution in T line of lateral-flow strip. A) Optical images show the time-dependent flow of F-BSA solution in the CA band. Scale bar = 0.5 mm. B) The plot shows the fluorescence intensity of F-BSA in CA band with time.

We showed the flow behavior of the sample solution on the lateral-flow strip in Figure R10A. The sample solution could flow fast on the BAS Mem (for example, the distance from the conjugate pad to the T line), but slowed down at the T and C lines. Specifically, as shown in Figure R10B, the flow velocity on the BAS Mem reached 5.32 mm/s. In contrast, the flow velocity at the T line was only 0.25 mm/s, which was comparable to that on NC Mem. This indicates that our proposed lateral-flow strip provides sufficient time for “antigen-antibody-nanogold” standing in the detection area to complete the immunological detection.

Figure R10. The flow behavior of sample solution on BAS Mem for ultra-fast LFA. A) The schematic graph illustrates that the flow velocity is high on BAS Mem, but low at T line. B) The plot illustrates the variation in the flow velocity of liquid at different positions along the lateral-flow strips during the flow of the sample solution.

Meeting prerequisite 2: Additionally, it was important to note that the time for immunological binding between the “antigen-antibody-nanogold” complex and the antibody had a decisive impact on the detection time for LFA. If the time for immunological binding between the “antigen-antibody-nanogold” complex and the antibody was sufficient, it would be possible to increase the flow velocity of the liquid through the detection area (T line) without sacrificing the detection sensitivity and accuracy in the development of rapid lateral-flow strips. With the increase in the flow velocity of the liquid, we can thereby reduce the overall detection time.

The immunological binding between the “antigen-antibody-nanogold” complex and the antibody in T line was primarily influenced by the time taken for the free diffusion of “antigen-antibody-nanogold” complex to the antibody immobilized at the T line (which was fixed to the surface of the CA fiber). According to Fick's law and the Einstein equation, we got:

$$\langle x^2 \rangle = 6DT$$

Herein, x represented the distance of free diffusion of sample molecules or nanoprobe (such as “antigen-antibody-nanogold” complex), D was the diffusion coefficient of “antigen-antibody-nanogold” complex, and T was the time of free diffusion of “antigen-antibody-nanogold” complex. To determine the distance of free diffusion of sample molecules in T line, we characterized the T line which was basically the cellulose acetate band (CA band) in Figure R11. The results showed that the CA band was embedded the surface structure of the BAS Mem (Figure R11A), and featured a large number of pores internally (Figure R11B). We approximately set the distance of free diffusion of sample molecules as 2.605 μm (which was the half of the average pore size of CA band), according to the Figure R11C.

Figure R11. Characterization of the shape, microstructure, and pore size distribution of CA bands used as T and C line. A) Optical images showing the structure of lateral-flow strips and CA bands. B) SEM images showing the internal structure of the CA bands. C) Histogram of pore size distribution of CA bands.

Therefore, we determined that it took approximately 0.25 s for the “antigen-antibody-nanogold” complex to freely diffuse to the immobilized antibody in the T line through the calculation of the above equation. Considering it took 8.00 s for the sample solution carrying the target sample molecules to pass through the T line (Figure R9). Consequently, there is sufficient time for the “antigen-antibody-nanogold” complex to immunological binding with the antibody in T line at the current flow velocity of liquid (Figure R12).

Time for  and  immunological binding in T line: **0.25 s**

Time for  flow through T line: **8.00 s**

 antigen  antibody  antigen-antibody-nanogold

Figure R12. The schematics shows the time is sufficient for the immunological binding between the “antigen- antibody- nanogold” complex and the antibody immobilized at the T line.

3) After meeting the two prerequisites mentioned-above, we could achieve highly sensitive detection. The enhancement of detection sensitivity mainly relied on two aspects as follow: First, we enabled more sample solution to flow toward and flow through the T and C lines, and increased the signal the T line (Figure R13C). Second, we reduced the residue of the sample solution, the non-specific adsorption of target protein molecules, and the retention of antibody-labeled signal amplification nanoprobes, so that decreasing the background noise (Figure R13C). Consequently, our lateral-flow strip significantly improved the SNR of lateral-flow strip (Figure R13D), and thereby enhancing the detection sensitivity of LFA.

Figure R13. Results of lateral-flow strips constructed with NC Mem and BAS Mem for the assay of positive sample. A) Optical images showing the lateral-flow strips that constructed with NC Mem and BAS Mem, respectively. Scale bar = 5 mm. B) The plot showing the variation in color intensity for lateral-flow strips that constructed with NC Mem and BAS Mem, respectively. C) Histogram showing the increase of signal and the decrease of noise using lateral-flow strips constructed by BAS Mem, compared to NC Mem. D) Histogram showing the increase of SNR by using lateral-flow strips constructed by BAS Mem, compared to NC Mem.

In conclusion, the reasons why our proposed lateral-flow strip could transport liquid at an increased velocity, while maintaining a low limitation of detection (LOD) were that: We made the

flow velocity was high on BAS Mem, but low at T line, and we ensured the time was sufficient for the immunological binding between “antigen-antibody-nanogold” complex and antibody. Building on these two prerequisites, we increased the signal at T line and decreased the noise in non-detectable area by using BAS Mem. Therefore, we increased the SNR of lateral-flow strip, which in turn enhanced the sensitivity of LFA

In order to address why the flow velocity has increased while the LOD also performs well, we have revised the details of the characteristics of lateral-flow strip in the **Results** section of revised the manuscript accordingly. “Fig.1D conceptually illustrates the flow of liquid on the lateral-flow strips. It showed that the flow velocity was high on the BAS Mem of our lateral-flow strip, whereas it was lower at the T line. The reduced flow velocity at the T line resulted in a longer residence time of the liquid at this location. This extended residence time was crucial, as it allowed sufficient time for the immunological binding between the “antigen-antibody-nanogold” complex and the antibody immobilized in the T line.” (See page 8, line157-161)

To make the mechanism underlying the ultra-fast and highly sensitive LFA clearer, we have added detailed description into the **Discussion** section of revised the manuscript. “Firstly, the flow velocity of liquid was high on BAS Mem, but low at T line (comparable to that on NC Mem) (Supplementary Fig. 19). Then, the time was sufficient for the immunological binding between “antigen-antibody-nanogold” complex and antibody in the T line of our proposed lateral-flow strip (Supplementary Fig. 20-21 and Supplementary Note 4). Building on these prerequisites, we increased the signal at T line and decreased the noise in non-detectable area by using BAS Mem. The SNR of lateral-flow strip was increased (Supplementary Fig. 16), so that enhanced the detection sensitivity of LFA.” (See page 18, line 361-366)

Comment 3: *Another thing should taken into consider is the surface to volume ratio (SVR) of the substrate, which also contribute a lot to higher sensitivity. I don't think the SVR is good for the quasi-3D BAS in sub millimeter scale in comparison to the porous 3D structure of NC Mem. In this case, I think one point is very important. The authors should clarify why cellulose acetate (CA) is used to draw the T and C lines. What's the microscopic structure of the actual T or C line, for example by SEM? And what's the actual flow velocity is at this kind of T line structure?*

Response: Thank you for the comment. We would like to discuss the comments as follows.

1) Yes, the surface to volume ratio (SVR) of the substrate would also contribute a lot to the detection sensitivity. Therefore, we introduced the additional components with a three-dimensional porous structure on the BAS Mem to immobilize antibodies. As shown in Figure R11A of **comment 2**, the additional components were two bands made of cellulose acetate (CA bands), and were used to serve as the T and C lines of lateral-flow strip. The CA band was embedded in the surface structure of the BAS Mem. The scanning electron microscope (SEM) image in Figure R11B displayed the microscopic structure of the T line, showing there were large number of pores in the CA band. Figure R11C showed the pore size distribution graph (data were obtained from Figure R 11B), and the average pore size of CA band was 5.21 μm . These additional components (CA band) helped to increase the surface area available for immobilization of antibody, and could shorten the diffusion distance required for the immunological binding. This structural design ensures that surface area on the BAS Mem is no longer a limiting factor for antigen-antibody binding.

Figure R11. Characterization of the shape, microstructure, and pore size distribution of CA bands. A) Optical images showing the structure of lateral-flow strips and CA bands. B) SEM image showing the internal structure of the CA bands. C) Histogram of the pore size distribution of CA bands.

2) There are several reasons why cellulose acetate (CA) is used to draw the T and C lines:

Firstly, this CA material is easily fabricated into porous fibrous structures, providing enough space for the immobilization of antigens and antibodies, and meets the requirements for constructing T and C lines in LFA.

Secondly, the chemical structure and surface properties of CA make it less likely to non-specifically adsorb the antigens and antibodies, which helps to reduce background noise, and enhancing the SNR of the LFA.

Additionally, CA material performs good biocompatibility, chemical stability, and durability, making it widely suitable for various biological applications, particularly due to its strong tolerance and stability against changes in pH values and different types of solvents.

Finally, the ease of processing CA material allows CA could be fabricated into products of various shapes and sizes through conventional processing techniques easily (such as extrusion, injection molding, and blow molding) at a relatively low cost.

In summary, material of CA provides multiple advantages in the preparation of T and C lines, which meet the requirements of the proposed lateral-flow strips.

In order to explain why cellulose acetate (CA) is used to draw the T and C lines, we added the information in the **Methods** section of the revised manuscript. “We used cellulose acetate (CA) material to prepare the BAS Mem with T and C lines, taking into account the multiple advantages of CA in the preparation of lateral-flow strips (Supplementary Note 7).” (See page 22, line 450-451)

3) We provided additional SEM images to exhibit microscopic structure of the actual T and C line in Figure R11B. The results indicated that the T line in our lateral-flow strip was essentially made of the three-dimensional porous fiber material (referred to CA).

Furthermore, we supplemented experiments in Figure R9, showing the actual flow velocity at this kind of T line structure. Based on this, we could calculate that the average flow velocity of the liquid in the T line was about 0.25 mm/s.

To exhibit microscopic structure of the actual T or C line, we have supplemented with a scanning electron microscope (SEM) image of the T and C lines, as shown in the Supplementary Fig. 20 in supporting information of revised manuscript.

To show the actual flow velocity at this kind of T line structure, we have supplemented the

experiments by measuring the flow velocity of F-BSA solution in the T line, as shown in Supplementary Fig. 19A-B in the supporting information of revised manuscript.

Comment 4: *In order to obtain contact angles that meet the requirements, oxygen plasma treatment is selected to obtain different contact angles for further analysis. What is the duration of hydrophilicity. Usually, it is not very stable and then this will affect the shelf time of the strip. Then, what kind of hydrophilic coating was chosen for use here?*

In general, I do not think the novelty and significance of the manuscript meet the criteria of the journal.

Response: Thank you for your comment.

1) Yes, the surface hydrophilicity improved by the oxygen plasma treatment could not maintain for a long-term duration. Therefore, when BAS Mem was used to construct lateral-flow strips, the surface hydrophilicity of the BAS Mem needed to be achieved through the application of hydrophilic coatings, rather than through the oxygen plasma treatment. Herein, the purposes of the oxygen plasma treatment were: enhancing the stability and uniformity of the hydrophilic coating on the surface of BAS Mem, and improving the adhesion of subsequent hydrophilic coating.

Figure R14A demonstrated the preparation procedure of surface hydrophilicity treatment on the BAS Mem in an actual preparation. First, we modified the surface of the BAS Mem with oxygen plasma treatment for 30 s. Subsequently, we applied the hydrophilic coating onto the surface of plasma-treated BAS Mem by spraying, and continuing for 30 s. Finally, we let the BAS Mem stand at room temperature for 30 s to dry the hydrophilic coating on the surface, thus completing the surface treatment for its hydrophilic properties.

Figure R14. Preparation procedure for the surface treatment of the BAS Mem, and the hydrophilicity of BAS Mem that stored for various durations. A) Schematic graphs showing the steps for preparing the hydrophilized BAS Mem. B) Optical images showing the flow behavior of liquid on BAS Mem that stored for various durations. C) Histogram showing the flow velocity of liquid on BAS Mem that stored for various durations. D) The plot showing the relationship between the flow velocity of liquid on BAS Mem and the stored durations.

The BAS Mem prepared by the above method was able to maintain long-term hydrophilicity. To show this, we supplemented the experiment for the long-term effectiveness of hydrophilicity on the

surface of BAS Mem. We took BAS Mem stored for 0, 2, 4, 12, and 24 weeks, applied 10 μ L of red ink, and observed the flow behavior of liquid on different membranes. As shown in Figures R14B-D, the flow velocity of the liquid on the BAS Mem stored for 24 weeks was 5.40 mm/s, and the flow velocity on the BAS Mem stored for 0 weeks was 5.51 mm/s. The results show that the hydrophilicity of the BAS Mem prepared by this method could last for more than six months, which meant that the BAS Mem would not affect the shelf life of the lateral-flow strip.

To show the role of the oxygen plasma treatment for the preparation of BAS Mem clearer, we have added the statement and graph for the preparation procedure of the BAS Mem in the **Methods** section of revised the manuscript. “Finally, we removed the HDPE membrane from assembly and modified the surface of HDPE membrane with a hydrophilic coating according to the procedure outlined in Supplementary Fig. 24, and obtained the BAS Mem.” (See page 21, line 429-431)

To show the duration of hydrophilicity of BAS Mem, we supplemented the experiment and added the statement in the **Results** section of revised the manuscript. “The long-term hydrophilicity of the surface of BAS Mem had also been confirmed experimentally. The prepared BAS Mem maintained its hydrophilic properties consistently, and could transport the liquid effectively after a storage period of six months (Supplementary Fig. 11).” (See page 8, line 154-156)

2) The main component of the hydrophilic coating we used in practical applications was the prepolymer of acrylic resin. The hydrophilic coating contained a sufficient number of hydrophilic functional groups. The hydrophilic coating could bond well to the substrate. Besides, the hydrophilic coating also exhibited good wear resistance after being coated on the surface of the BAS Mem.

To make this clearer, we have added statements in the **Methods** section of revised manuscript. “The composition of the hydrophilic coating, and the methods for controlling its hydrophilicity were presented in Supplementary Note 6.” (See page 22, line 436-437)

3) Finally, we would like to discuss novelty and significance of the manuscript, by discussing the degree of advances, depth of theoretical part and the impact of this work as follow:

In terms of the advanced innovation of our work, firstly, we proposed the novelty BAS Mem, which **exhibited the highest rectification coefficient that reported to date**. The BAS Mem enabled the unidirectional, rapid, and low-residual flow of liquid. Secondly, the BAS Mem, which transports liquid through surface microchannels, offers a greater variety of sample solution types, a wider range of sizes for testable sample molecules and antibody-labeled signal amplification nanoprobe in LFA, compared to traditional fiber chromatographic membranes (such as NC Mem). **This feature significantly broadens the applicability of the LFA, especially for those larger sample molecules and nanoprobe that are challenging to process with traditional NC Mem.** As shown in Figure 15, based on the detection that used 500 nm- nanospheres as antibody-labeled signal amplification nanoprobe, we further utilized larger-sized microspheres (diameter 1 μ m) as antibody-labeled signal amplification probes and applied them on the strips constructed with both NC Mem and BAS Mem. The results showed that the lateral-flow strips made with BAS Mem successfully completed qualitative detection, whereas those made with NC Mem did not (Figure R15E). The lateral-flow strips made with BAS Mem showed a 541.8% increase in the detection signal at the T line, and an 84.7% reduction in noise compared to the strips made with NC Mem (Figure R15G). The SNR of lateral-flow strip was 19.2, which was 32 times higher than that of the strips made with NC Mem (Figure R15H). The BAS Mem provides a broader selection space for sample solutions and antibody-

labeled signal amplification probes by facilitating fluid flow through its surface microchannels. The lateral-flow strip enables the detection of larger analyte molecules, and offers a solution to the volume limitations encountered with larger-sized antibody-labeled signal amplification probes during use. Thirdly, the compatibility of BAS Mem in the LFA opened up new application for surface flow membranes. **The lateral-flow strips constructed with BAS Mem can complete detection within 4 min, with a LOD of only 1.96 pg/ml, and achieving ultra-fast, high-sensitive home detection for cTnI.**

Figure R15. Results of lateral-flow strips constructed with NC Mem and BAS Mem for the assay of positive sample solution using different types of antibody-labeled signal amplification probe. A-D) Using 500 nm-gold particle as the antibody-labeled signal amplification nanoprobe. E-H) Using 1 μm-microsphere as the antibody-labeled signal amplification microprobe. A) and E) Optical images showing the lateral-flow strips that constructed with NC Mem and BAS Mem, respectively, with visible color on membranes. Scale bar = 5 mm. B) and F) The plot showing the variation of color intensity on lateral-flow strips constructed with NC Mem and BAS Mem, respectively. C) and G) Histograms showing the increase of signal and the decrease of noise by using lateral-flow strips constructed by BAS Mem, compared to NC Mem. D) and H) Histograms showing the increase of SNR by using lateral-flow strips constructed by BAS Mem, compared to NC Mem.

We conducted an in-depth study on the mechanism of fluid control, and proposed the principle of driving and pinning the liquid using the sidewalls of the BAS Mem. **We found that, the BAS Mem uses the sharp edges on the sidewalls of microchannels to prevent the wetting and spreading of liquid, and making use of the surface tension of liquid and capillary forces to driving the liquid, thereby achieving the unidirectional transport of liquid.** In addition to the special design of the sidewalls of the structure of BAS Mem, the contact angle of the surface of the BAS Mem was another key to realize mechanism of unidirectional flow of liquid. For the

unidirectional flow of liquid, we determined that the contact angle (θ) should lie between 48° and 64° . This mechanism of fluid control and principle can offer new ideas and methods for analyzing other fluid behaviors in the microfluidics field. Additionally, it may be widely applicable to solving technical problems related to fluid control.

Finally, our work covered a wide range of subjects, and could provide a new method for medical testing, and also has potential applications in multiple fields including materials chemistry, surface science, fluid dynamics, and public health. Here, we presented an example application of using capillary tubes prepared with BAS Mem for passive sampling in Figure R16. We rolled the BAS Mem that made of PP into a capillary tube with an inner diameter of 1.5 mm, and used the capillary tube for the unpowered vertical transport of liquid. Figure R16A showed the capillary tube prepared by flat membrane (flat Mem), and the performance of the vertical transport of liquid. After 40 s, the liquid had risen only about 5.0 mm inside the capillary tube prepared by flat Mem. In contrast, liquid in the capillary tube prepared by BAS Mem rose 14.0 mm up within 1 s (Figure R16B). After 40 s, the liquid had risen 21.0 mm along the capillary tube prepared by BAS Mem, which was 4.2 times higher than the height achieved by the capillary tube prepared by flat Mem. This demonstrated that the capillary tube prepared by BAS Mem could offer new approaches to the unpowered vertical transport of liquid. They could also be developed into one-way fluid transfer pipettes to avoid the contamination caused by the back-and-forth fluid movement in medical applications. Thus, our work not only provides a new method for medical diagnostics but also has potential applications in the fields of materials chemistry, surface science, fluid dynamics, and public health.

Figure R16. Capillary tubes prepared by flat Mem and BAS Mem, and the application of capillary sampling. A) Optical images showing the capillary tubes prepared by flat Mem, and time-dependent red ink rises in the capillary tube. B) Optical images showing the capillary tubes prepared by BAS Mem, and time-dependent red ink rises in the capillary tube.

To Reviewer #2:

Referee letter: Dear Author, this manuscript focused attention on the development of novel lateral-flow assay to improve its detecting velocity and efficiency. Use of unidirectional liquid transport mechanism on disease diagnosis is very interesting. The following points should be reconsidered.

Response: Thank you for the positive feedback and constructive suggestions.

Comment 1: Barbed arrow-like groove is designed to realize the unidirectional liquid transport. The test experiments also validate that this groove can enhance transport velocity and reduce the residue of sample solution. However, all these results are depended on the parameter of groove structure. It is very important to show how to determine and how to design.

Response: Thank you for the suggestion. We would like to discuss the comments as follows.

1) In order to determine the parameters of the groove structure, we referred to several structural parameters mentioned in the classical theoretical formulations. These structural parameters could affect the forces generated by sidewalls of the structure on the liquid. And we further combined them with the needs of practical applications for lateral-flow assay (LFA).

In our work, we primarily use the sharp edges on the sidewalls of microchannels to prevent the wetting and spreading of liquid, and making use of the surface tension of liquid and capillary forces to drive the liquid, thereby achieving unidirectional liquid transport. Therefore, in designing the barbed arrow-like structure (BAS), we paid special attention to the shape and size parameters of the BAS sidewalls. We adjusted the shape of the BAS sidewalls through four key variables: α represents the arc degree of the short arc sidewall, β represents the arc degree of the long arc sidewall. W represents the shortest width between the long arc sidewalls within the structural unit, and H represents the height of the sidewalls. In selecting these four parameters, we primarily considered several key factors including fluid dynamics, surface tension balance, and the interaction between fluid and structure.

2) The classical Gibbs inequality is not only very important in the study related to liquid surface tension and contact angle, but also has significant implications when discussing the wetting and spreading behavior of liquids on solid surfaces. The Gibbs inequality can be expressed as:

$$\theta - \theta_0 \leq 180^\circ - \varphi$$

Where θ is the actual contact angle of liquid on the surface, θ_0 is the equilibrium contact angle of the liquid, and φ is the angle of the sharp edge of sidewall. The Gibbs inequality reveals that whether a liquid, upon contacting a sharp edge (such as the edge of a microcavity), can be pinned (i.e., no longer continue to spread or slide) depends on its contact angle and the geometric shape of the edge. If this inequality is satisfied, then the liquid will be pinned at that sharp edge and cannot continue to spread beyond the edge. Thus, we know that in addition to the contact angle, the geometric shape of the edge is also an important factor affecting liquid behavior. Therefore, we chose to adjust the sharp corners of the BAS structure sidewalls as a parameter to modulate fluid behavior. We believe that by adjusting the angle of the sidewalls, we can control the behavior of liquids on the BAS Mem, effectively achieving liquid pinning. This discovery is significant for optimizing the BAS Mem structure and enhancing its performance in controlling liquid flow.

Many researchers have conducted comprehensive analyses of liquid flow behavior in classical

square capillary microchannels. For instance, Ichikawa et al. derived a formula for capillary force in square capillary microchannels (see 10.1016/j.jcis.2004.07.017). Their research indicates that the cross-sectional dimensions of the channel, namely width and height, significantly influence the capillary flow of liquid. Therefore, we selected the spacing (W) between the sidewalls and the height (H) of the sidewalls as two key parameters for adjusting the shape of the BAS structure. Therefore, we designed and optimized the arc degree of the short arc sidewall (α), and the arc degree of the long arc sidewall (β) under the guidance of the Gibbs inequality. And we determined the width (W) and height (H) of microchannels, guided by the flow behavior of liquid in square capillary microchannels.

Additionally, considering the practical application requirements and the specific demands for components of lateral-flow strip, we also included the spacing between each row of channels into our design parameters for detailed definition, represented by the letter S .

3) The design concept of the BAS Mem is shown in Figure R15. Initially, we noted the foundation that liquid could achieve capillary flow through capillary force in the square capillary microchannel membranes. According to the research by Ichikawa et al., the parameters of H and W could affect the capillary force, so that they were the key parameters of capillary flow. However, this type of surface structure membrane has limitations in directional transport and efficiency, and resulting in ordinary performance in flow velocity. Therefore, we introduced the symmetrical triangular prism blocks into the microchannels. The addition of blocks changes the geometric shape of the sidewalls, and creating the sharp edges on the channel sidewalls. According to the Gibbs inequality, we could make it difficult for the liquid to wet the sharp edges on the sidewalls in the backward flow direction, by adjusting the contact angle of surface of BAS Mem. Therefore, we prevented the reverse wetting and spreading of the liquid, and realized the unidirectional flow that facilitated by the sidewall. In further design improvement, we adjusted the structure and edge curvature of the prismatic blocks to create curved sidewalls. This design could reduce dead zones of fluid, turbulence, and flow resistance, thus enhancing the efficiency of liquid flow. All these design improvements aimed to enhance the performance of the BAS Mem in microfluidic applications, and ensuring the transport efficiency and directionality flow of liquid.

Figure R17. Schematic diagram showing the design concept of the BAS Mem.

To illustrate how to determine the parameter of groove structure and how to design the structure, we have added the information in the **Methods** section of the revised manuscript as follows. "We designed with patterns of barbed arrow-like structure based on the design concepts shown in Supplementary Fig. 23 and Supplementary Note 5, and used the techniques of laser carving, casting

and hot embossing to successively to prepare the BAS Mem.” (See page 21, line 421-423)

Comment 2: Oxygen plasma for different periods of time was used to adjust the surface wettability. This surface treatment can not remain long period. In practical application, how do you adjust surface wettability.

Response: Thank you for the question.

1) Yes, the surface wettability of BAS Mem, that treated with oxygen plasma for different periods of time, cannot remain long period. Therefore, the surface hydrophilicity of the BAS Mem needed to be achieved through the application of hydrophilic coatings, rather than through the oxygen plasma treatment. Figure R14A demonstrated the preparation procedure of surface hydrophilicity treatment on the BAS Mem in an actual preparation. First, we modified the surface of the BAS Mem with oxygen plasma treatment for 30 s. Subsequently, we applied the hydrophilic coating onto the surface of plasma-treated BAS Mem by spraying, and continuing for 30 s. Finally, we let the BAS Mem stand at room temperature for 30 s to dry the hydrophilic coating on the surface, thus completing the surface treatment for its hydrophilic properties.

Figure R14. Preparation procedure for the surface treatment of the BAS Mem and the hydrophilicity of BAS Mem that stored for various durations. A) Schematic graphs showing the steps for preparing the hydrophilized BAS Mem. B) Optical images showing the time-dependent flow of liquid on BAS Mem that stored for various durations. C) Histogram showing the flow velocity of liquid on BAS Mem that stored for various durations. D) The plot showing the relationship between the flow velocity of liquid on BAS Mem and the stored durations.

The BAS Mem prepared by the above method was able to maintain long-term hydrophilicity. To show this, we supplemented the experiment for the long-term effectiveness of hydrophilicity on the surface of BAS Mem (Figure R14 B-D). We took BAS Mem stored for 0, 2, 4, 12, and 24 weeks, applied 10 μ L of red ink, and observed the flow behavior of liquid on different membranes. As shown in Figures R14B-D, the flow velocity of the liquid on the BAS Mem stored for 24 weeks was 5.40 mm/s, and the flow velocity on the BAS Mem stored for 0 weeks was 5.51 mm/s. The results show

that the hydrophilicity of the BAS Mem prepared by this method could last for more than six months, which meant that the BAS Mem would not affect the shelf life of the lateral-flow strip.

2) Also, the surface hydrophilicity of BAS Mem was adjusted by the composition of hydrophilic coatings used in practical applications. Specifically, the hydrophilic coatings we selected contain a sufficient number of hydrophilic functional groups, and primarily composed of acrylate resin prepolymers. We could effectively control the hydrophilicity and hydrophobicity of the hydrophilic coating by adjusting the length and flexibility of molecular chain, molecular weight of the resin, as well as the types and proportions of additives, and thereby regulating the surface hydrophilicity of the BAS Mem. Besides, the characteristic of this hydrophilic coating is that it forms a strong bond with the substrate, and exhibits good abrasion resistance after application.

Therefore, the hydrophilic coating could form a uniform hydrophilic layer on the surface of BAS Mem, so that the BAS Mem could perform a long period of hydrophilicity with this surface treatment.

To show this hydrophilic treatment using hydrophilic coatings on the BAS Mem can remain effective for a long period, we supplemented the experiment and added the statement in the **Results** section of revised the manuscript. “The long-term hydrophilicity of the surface of BAS Mem had also been confirmed experimentally. The prepared BAS Mem maintained its hydrophilic properties consistently, and could transport the liquid effectively after a storage period of six months (Supplementary Fig. 11).” (See page 8, line 154-156)

In order to show the method of adjusting the surface wettability of BAS Mem in practical application, we have added statements in the **Methods** section of revised manuscript. “The composition of the hydrophilic coating, and methods for controlling its hydrophilicity were presented in Supplementary Note 6.” (See page 22, line 436-437)

Comment 3: *It will much more better to define unidirectional liquid transport coefficient than just equation derivation as like Eq. (1)-(2).*

Response: Thank you for the suggestion. We defined the rectification coefficient as the unidirectional liquid transport coefficient in our work. The rectification coefficient is defined as the ratio of the distances, that the liquid spreads to both sides from the drop position. The rectification coefficient is a widely used parameter for evaluating the performance of unidirectional liquid flow in the field of unidirectional fluid control.

To make this clearer, we have added a detailed definition of the liquid transport coefficient and designated this definition as *Eq. (3)* in the **Results** section of revised manuscript. “The rectification coefficient of liquid flow on membranes, which was termed “k”, could express as (3).”

“
$$k = \frac{L_s}{L_p} \tag{3}$$

Herein, L_s represented the lengths of liquid forward flowing, and L_p represented the lengths of liquid backward flowing.” (See page 6, line 116-119)

Comment 4: *This manuscript shows one practical application in disease diagnosis. All the line or parameter setting should consider the test demand or practical application. For example, how to determine the T line and C line in Fig.3.*

Response: Thank you for the suggestion.

1) Lateral-flow assay (LFA) is a crucial tool for on-site and real-time diagnosis. The LFA strips are portable and easy to operate, so that they are particularly suitable for epidemiological screenings and home testing. Currently, the LFA strips typically require 15-30 min to produce results of the immunological detection. Moreover, the sensitivity of LFA strips is generally low, making it difficult to reliably detect samples at low-concentration. These limitations in detection time and sensitivity restrict their use in clinical applications. For instance, in cases of diseases that may present sudden attacks and pose threats to life and health, such as acute myocardial infarction (AMI), LFA strips struggle to rapidly detect early, low-concentration biomarkers on-site, thereby hindering timely diagnosis for potential AMI patients (such as those exhibiting symptoms like chest pain). Additionally, doctors need to quickly ascertain information about certain low-concentration blood biomarkers in emergency medical situations. However, the current LFA are unable to accomplish these detections. Therefore, there is an urgent need to develop an LFA method that can achieve higher sensitivity in a shorter time.

In response to the need for rapid and high-sensitivity capabilities in LFA strips, we proposed the BAS Mem, that enabled the unidirectional flow of liquid, and used this BAS Mem, instead of the most widely used fiber-based chromatography membrane (i.e., NC Mem), to construct a novel type of lateral-flow strip. The lateral-flow strip could realize the ultra-fast and highly sensitive LFA. The BAS Mem had excellent performance in driving sample solution unidirectionally when it was used as a component to construct the lateral-flow strip for ultra-fast hs-cTnI assay, as listed as follows. (1) The flow time of liquid on the BAS Mem approximately 93.5% lower than that on the NC Mem, indicating a shorter flow time of liquid on the BAS Mem compared to the NC Mem. (2) By increasing the utilization rate of the sample solution, the BAS Mem enabled an efficient flow of sample solution, transporting a greater amount of detectable complex to the T line, so as to increase the detection signal. (3) The residual sample solution on the BAS Mem was reduced by 48% compared to that in the NC Mem, minimizing the background noise in non-detectable areas. Consequently, the BAS Mem in lateral-flow strips could be used to reduce the detection time and enhance the sensitivity of LFA. The results indicated the detection performance, including (1) a shortened detection time of 4 min, and (2) a lowered limit of detection (LOD) to 1.97 pg/mL. The LFA based on the proposed lateral-flow strips achieved a specificity of 100% and a sensitivity of 93.3% in the detection of the serum samples of 25 suspected AMI patients. Compared to the performance of other commercial-available cTnI strips, the proposed lateral-flow strips saved 66.7%- 86.7% of the detection time, and achieved a LOD that was lowered by 3 - 5 orders of magnitude with a low cost.

2) In terms of the determination of the location of the T and C lines on the lateral-flow strips (e.g., Figure 3D, etc.), we referred to the industry standards that are widely used at present. We have aligned the positions of the T and C lines on our lateral-flow strips with those of standard strips commonly found on the market. The dimensions of each component of the lateral-flow strip are consistent with the existing strips available on the market. We provided a detailed assembly schematic in Figure R18A, and also marked the dimensions for each component.

Besides, to demonstrate how to observe the signal at the T and C lines on our lateral-flow strips more clearly, we provided an actual photo of the lateral-flow cassette in Figure R16B. The lateral-flow strips constructed with BAS Mem are encased in a plastic casing. The lateral-flow cassette is 75 mm in length and 20 mm in width. The lateral-flow cassette featuring a sample hole, and test and

control lines marked with “T” and “C”, respectively.

Figure R18. Characterization of the lateral-flow strip made of BAS Mem. A) Optical image showing the structure of the lateral-flow strip, and the dimensions of each component. B) Optical image shows the physical layout of a lateral-flow cassette.

To avoid any misunderstanding about the positions of the T and C lines in Figure.3, we have added an explanation in the figure caption. “The black dashed lines marked in the image indicate the location of practical T and C lines.” (See page 14, line 267-268)

Also, to make the determination of the T and C lines clearer, we have revised statements in the **Methods** section of revised manuscript. “Finally, we assembled all the components into a lateral-flow strip according to the Supplementary Fig. 25A.” (See page 23, line 469) “The assembled lateral-flow strip was then encapsulated within a white plastic casing. The lateral-flow cassette measures 75 mm in length and 20 mm in width. The center of the lateral-flow cassette features a hollowed-out area equipped with a sample hole, as well as designated test and control regions marked with “T” and “C”, respectively (Supplementary Fig. 25B).” (See page 23, line 473-476)

Comment 5: *The organization of this manuscript should be polished.*

Response: Thank you for the suggestion. In order to polish the organization of this manuscript, we have revised the “**Results**” section by re-organizing Figure 1, Figure 3, and their description to optimize the presentation of information and enhance logical coherence. Besides, we also have revised the “**Discussion**” section by adding statements and supplemented experiment to display the strategies and innovations of our work more comprehensively and clearly. The details are as follows.

1) In the “**Results**” section

(1) In order to more visually demonstrate the flow of the liquid on our lateral-flow strip, and details the immunobinding at T line, we adjusted the schematic diagram of the flow of the liquid on the BAS Mem (FigureR19), and modified the statements in the **Results** section of the revised manuscript accordingly. “Fig.1D conceptually illustrates the flow of liquid on the lateral-flow strips. It showed that the flow velocity was high on the BAS Mem of our lateral-flow strip, whereas it was lower at the T line. The reduced flow velocity at the T line resulted in a longer residence time of the

liquid at this location. This extended residence time was crucial, as it allowed sufficient time for the immunological binding between the “antigen-antibody-nanogold” complex and the antibody immobilized in the T line.” (See page 8, line157-161)

The adjusted Figure 1D can more clearly show the flow characteristics of the liquid on the lateral-flow strip: The liquid flows fast on the BAS Mem but slow at the T line. And the time will sufficient for the immunological binding in the T line of lateral-flow strip. Moreover, avoiding the misinterpretation that rapid detection may lead to ineffective capture in the T line.

Figure R19. (i.e., **Figure 1**) Design and structure characterization of the BAS Mem, the principle of unidirectional flow of liquid on BAS Mem, the flow of liquid on a mimic strip and

the application of BAS Mem to construct lateral-flow strips for ultra-fast hs-cTnI assay. (A) Design and structure characterization of the BAS Mem. A-i) Optical image of the BAS Mem. Scale bar= 2 mm; A-ii) Magnified SEM image of the BAS Mem. Scale bar= 500 μm ; A-iii) Top-SEM view of the BAS Mem. Here, α denoted the degrees of the short arc on the short-arc sidewalls, and β denoted the degrees of the long arc on the long-arc sidewalls, respectively. W denoted the shortest width between the long-arc sidewalls within a structure unit. Scale bar= 500 μm ; A-iv) SEM image of cross-sectional view of the BAS Mem. H denoted the heights of the sidewalls. Scale bar= 500 μm . (B) Schematic drawings indicate the analysis of forces on the convex menisci when liquid was pinned on the BAS Mem, including the forces generated by the underside (left) and that by sidewall (right) respectively. (C) Schematic drawings indicate the analysis of forces on the concave menisci when water was spreading on the BAS Mem, including the forces generated by the underside (left) and that by sidewall (right) respectively. (D) Conceptual graph showing the characteristics of the lateral-flow strip made of BAS Mem, highlighting that the flow velocity is high on BAS Mem but low at T line, and the time is sufficient for the immunological binding in T line. (E) Application of BAS Mem for ultra-fast lateral-flow assay of cTnI. The schematic drawing shows the process of detecting cTnI in peripheral blood within 4 min using BAS Mem-based lateral-flow strips for diagnosis of chest pain that occurs in humans.

(2) To more clearly demonstrate that our BAS Mem has a higher rectification coefficient compared to NC Mem, and advantages which this increased rectification coefficient brings to the LFA, we re-organized the order of descriptions in the **Results** section of the revised manuscript as follows. “We compared BAS Mem with the prevalent nitrocellulose membranes (NC Mem), a typical example of fiber-based chromatographic membranes, in terms of the flow directionality of liquid, flow time of liquid, utilization rate of liquid, and residual of sample solution on the membrane.” (See page 11, line 211-213) “First, we investigated directionality of liquid flow on the membranes.” (See page 11, line 214) “Then we investigated the flow time of liquid on membranes.” (See page 11, line 222) “We further investigated the utilization rate of the sample solution when flowing on BAS Mem.” (See page 12, line 233)

To more clearly demonstrate that our BAS Mem has a higher rectification coefficient compared to NC Mem, we revised the statements and histograms in Figure 3A-ii and Figure 3A-iii. “Fig. 3A-ii presented the bar graph comparing the length of forward and backward liquid flowing in millimeters. The liquid on the BAS Mem exhibited a forward flow of 43.0 mm, and a backward flow of merely 3.0 mm. In contrast, the lengths of liquid flow in both directions on the NC Mem were relatively similar. As shown in Fig. 3A-iii, the rectification coefficient for BAS Mem reached 14.5, which is 13.5 times higher than that of the NC Mem (1.02).” (See page 11, line 216-220)

To visually compare the BAS Mem can help to reduce the detection time of LFA, we revised the content of the comparison: we changed the comparison of flow velocity of liquid on BAS Mem and NC Mem to the comparison of the flow times of liquid on BAS Mem and NC Mem in Figure 3C. “The flow time of liquid on the BAS Mem approximately 93.5% lower than that in the NC Mem (Fig.3C), indicating a shorter flow time of liquid on the BAS Mem compared to the NC Mem.” (See page 11, line 226-227) “These results suggested the BAS Mem enabled shorter flow time of liquid, comparing to the NC Mem. Consequently, the BAS Mem has the potential to shorten the time of sample solution taken to flow to the T and C lines on lateral-flow strips, and has contributed to the

reduction of detection time in LFA.” (See page 12, line 229-232) And we moved the histogram, showing flow velocity of liquid on BAS Mem and NC Mem, into the revised supporting information. “Further calculations revealed that the average flow velocity on the BAS Mem was 5.32 mm/s, which was approximately 15.6 times faster than in the NC Mem (0.32 mm/s) (Supplementary Fig. 14).” (See page 11, line 227-229)

In order to more visually compare the usage of sample using BAS Mem and NC Mem, we have added the histogram in Figure 3E. “The usage of sample on the BAS Mem approximately 70% lower than that in the NC Mem (Fig.3E), indicating a higher utilization rate of sample solution on the BAS Mem compared to the NC Mem.” (See page 12, line 239-241)

Figure R20. ((i.e., Figure 3) Characterization of the performance of BAS Mem when they

were used to drive liquid and sample solution, including flow directionality of liquid, flow time of liquid, utilization rate of liquid, and residual sample solution on the membrane. (A) Characterization of flow behavior of liquid on two types of membranes. A- i) Optical images show the flow behavior of liquid on two types of membranes. Scale bar = 5 mm.; A- ii) Comparison histogram shows the lengths of liquid flowing on two types of membranes; A- iii) Comparison histogram of the rectification coefficients between two types of membranes. (B) Optical images show the time-dependent flow of liquid on two types of membranes. Scale bar = 5 mm. (C) Comparison histogram shows the time of liquid flowing on two types of membranes. (D) Optical images show the flow of liquid in different volumes on two types of membranes. The black dashed lines marked in the image indicate the location of practical T and C lines. Scale bar = 5 mm. (E) Comparison histogram shows the lengths of 5 μ L of liquid flowing on two types of membranes. (F) Optical images show the time-dependent flow of sample solution on two types of membranes, when the sample solution of Au-Ab was transported using mimic strips. Scale bar = 5 mm. (G) Curves show the time-dependent color intensity of Au-Ab on membranes. (H) Comparison histogram show the color intensity of Au-Ab that remained on BAS Mem and NC Mem (at 300 s).

The redrawn schematic figures are shown in Figure 3 (Figure R20). This Figure 3 shows more clearly about the rectification effect of the BAS Mem for liquid, and the advantages of the rectification effect on flow time, usage of sample and residue of sample solution. These polished demonstrations help to better understand the advantages of BAS Mem in controlling flow direction and efficiency.

2) In the “**Discussions**” section

(1) In order to address the mechanism underlying the ultra-fast and highly sensitive LFA clearer, we have added detailed description into the **Discussion** section of revised the manuscript. “**Firstly, the flow velocity of liquid was high on BAS Mem, but low at T line (comparable to that on NC Mem) (Supplementary Fig. 19). Then, the time was sufficient for the immunological binding between “antigen-antibody-nanogold” complex and antibody in the T line of our proposed lateral-flow strip (Supplementary Fig. 20-21 and Supplementary Note 4). Building on these prerequisites, we increased the signal at T line and decreased the noise in non-detectable area by using BAS Mem. The SNR of lateral-flow strip was increased (Supplementary Fig. 16), so that enhanced the detection sensitivity of LFA.**” (See page 18, line 361-366)

The modifications can more clearly explain the strategy of the proposed lateral-flow strip for ultra-fast and highly sensitive LFA.

(2) In order to more clearly and comprehensively demonstrate the innovative advantages of our work, and the possible broad applications. We have supplemented the experiments using colored spheres as antibody-labeled signal amplification probe and added the corresponding representations. “**The BAS Mem, which transports liquid on the surface, offers a greater variety of sample solution types, a wider range of sizes for testable sample molecules and antibody-labeled signal amplification nanoprobes in LFA, compared to traditional fiber-based chromatographic membranes (such as NC Mem). This feature significantly broadens the applicability of the LFA, especially for those larger target biomarkers (such as certain virus particles or polysaccharides) and antibody-labeled signal amplification probe (such as gold particles and magnetic bead particles) that are challenging to process with traditional NC Mem. To verify this, we used the larger-diameter nanospheres (with a**

diameter of 500 nm) as the antibody-labeled signal amplification nanoprobe, and conducted the detection using two types of strips (Supplementary Fig.22 A). The strips made of NC Mem failed to complete the assay with higher background noise than the detection signal (Supplementary Fig. 22 B). In contrast, the lateral-flow strips constructed with BAS Mem showed an increased signal at T line of 106.8%, and a decreased background noise of 79.0% compared to the strip constructed with NC Mem (Supplementary Fig.22 C). The SNR of lateral-flow strip made of BAS Mem was 11.7, which was 9.8 times higher than that of the strip made of NC Mem (Supplementary Fig.22D). The difference in performance of SNR between the strips constructed with NC Mem and BAS Mem became more pronounced when larger-sized antibody-labeled signal amplification probes were used. We further utilized larger-sized microspheres (diameter 1 μm) as antibody-labeled signal amplification probes and applied on the strips constructed with both NC Mem and BAS Mem. The results showed that the lateral-flow strips made with BAS Mem successfully completed qualitative detection, whereas those made with NC Mem did not (Supplementary Fig.22 E-F). The lateral-flow strips made with BAS Mem showed a 541.8% increase in the detection signal at the T line, and a 84.7% reduction in noise compared to the strips made with NC Mem (Supplementary Fig.22 G). The SNR of lateral-flow strip was 19.2, which was 32 times higher than that of the strips made with NC Mem (Supplementary Fig.22 H). Therefore, the BAS Mem provides a broader selection space for target biomarkers and antibody-labeled signal amplification probes by facilitating fluid flow through its surface microchannels. The lateral-flow strip enables the detection of larger size of biomarker, and offers a solution to the retention larger-sized antibody-labeled signal amplification probes that encountered recently.” (See page 19, line 380-402)

REVIEWERS' COMMENTS

Reviewer #1 (Remarks to the Author):

The reply is satisfactory and I think the manuscript is acceptable.

Reviewer #2 (Remarks to the Author):

This manuscript focus on the design of ultra-fast, highly-sensitive lateral-flow assay of cTnI, which is very interesting for readers. A barbed arrow-like structure membrane (BAS Mem) with ultra-high rectification coefficient which enables ultra-fast, highly-sensitive lateral-flow assay of cTnI was proposed. It will be much better to describe the design of barbed arrow-like structure in detail.

A point-by-point response to the reviewers' comments

To Reviewer #1:

Referee letter: The reply is satisfactory and I think the manuscript is acceptable.

Response: Thank you for the nice feedback.

To Reviewer #2:

Referee letter: This manuscript focus on the design of ultra-fast, highly-sensitive lateral-flow assay of cTnI, which is very interesting for readers. A barbed arrow-like structure membrane (BAS Mem) with ultra-high rectification coefficient which enables ultra-fast, highly-sensitive lateral-flow assay of cTnI was proposed. It will be much better to describe the design of barbed arrow-like structure in detail.

Response: Thank you for the nice feedback and suggestion. We have added more detailed information to describe the design of barbed arrow-like structure in Supplementary Methods 1 as follow.

“The design concept of the BAS Mem is shown in Supplementary Fig. 23. Initially, we noted the foundation that liquid could achieve capillary flow through capillary force in the square capillary microchannel membranes. According to the research by Ichikawa et al., the parameters of H and W could affect the capillary force, so that they were the key parameters of capillary flow. However, this type of surface structure membrane has limitations in directional transport and efficiency, and resulting in ordinary performance in flow velocity. Therefore, we introduced symmetrical triangular prism blocks into the microchannels. The height of these blocks was the same as the depth of the square capillary microchannel. The addition of these blocks changed the geometric shape of the sidewalls, which also resulted in the change of region for liquid flow. When the liquid flowed through the blocks, it could be found that the angle (φ) between the neighboring two sidewalls of the blocks could primarily influence flow behavior of liquid. By adjusting φ to a small acute angle, we created the sharp edges on the channel sidewalls. According to the Gibbs inequality, we could make it difficult for the liquid to wet the sharp edges on the sidewalls in the

backward flow direction, by adjusting the contact angle of surface of BAS Mem. Therefore, we prevented the reverse wetting and spreading of the liquid, and realized the unidirectional flow that facilitated by the sidewall. In further design improvement, we adjusted the structure and edge curvature of the prismatic blocks to create curved sidewalls. This design could reduce dead zones of fluid, turbulence, and flow resistance, thus enhancing the efficiency of liquid flow. All these design improvements aimed to enhance the performance of the BAS Mem in microfluidic applications, and ensuring the transport efficiency and directionality flow of liquid.” (The statement we added has been highlighted in blue, page 11, line 14-20 in supplementary information file.)